# A Theoretical Framework for Statistical Evaluability of Generative Models

**Shashaank Aiyer** [1]   **Yishay Mansour** [2 3]   **Shay Moran** [4 3]   **Han Shao** [1]

## Abstract

Statistical evaluation aims to estimate the generalization performance of a model using held-out IID test data sampled from the ground truth distribution. In supervised learning settings such as classification, performance metrics such as error rate are well-defined, and test error reliably approximates population error given sufficiently large datasets. In contrast, evaluation is more challenging for generative models due to their open-ended nature: it is unclear which metrics are appropriate and whether such metrics can be reliably evaluated from finite samples.

In this work, we introduce a theoretical framework for evaluating generative models and establish evaluability results for commonly used metrics. We study two categories of metrics: test-based metrics, including integral probability metrics (IPMs), and Rényi divergences. We show that IPMs with respect to any bounded test class can be evaluated from finite samples up to multiplicative and additive approximation errors. Moreover, when the test class has finite fat-shattering dimension, IPMs can be evaluated with arbitrary precision. In contrast, Rényi and KL divergences are not evaluable from finite samples, as their values can be critically determined by rare events. We also analyze the potential and limitations of perplexity as an evaluation method.

Authors are ordered alphabetically.

[1]Department of Computer Science, University of Maryland, College Park, USA [2]School of Computer Science, Tel Aviv University, Tel Aviv, Israel [3]Google Research, Tel Aviv, Israel [4]Faculty of Mathematics, Technion, Haifa, Israel. Correspondence to: Shashaank Aiyer <saiyer1@umd.edu>, Han Shao <hanshao@umd.edu>.

*Proceedings of the 43rd International Conference on Machine Learning*, Seoul, South Korea. PMLR 306, 2026. Copyright 2026 by the author(s).

## 1. Introduction

Leading approaches to the evaluation of generative models can be categorized into statistical evaluation (e.g., calculating scores such as perplexity and next-token accuracy on held-out data), benchmarks, which aggregate performance across multiple tasks (Liang et al., 2023; Jimenez et al., 2024), and human feedback (Chiang et al., 2024). While benchmarks provide standardized and interpretable comparisons, they assess performance on a fixed set of tasks, making it difficult to reason about generalization beyond the benchmark. Human feedback, in contrast, requires costly human labor.

By statistical evaluation, we mean assessing models using held-out IID data drawn from a ground truth distribution. Statistical evaluation is often used to evaluate pre-trained models and in settings where downstream tasks are unclear, such as synthetic data generation. Despite its relative simplicity and its inability to capture certain qualitative aspects of model behavior, statistical evaluation remains the simplest, cheapest, and most scalable way to compare models and offers a complementary alternative that directly targets generalization to the underlying data distribution. Unlike supervised learning, where population error is the gold-standard metric and test error is a reliable proxy for it, generative-model evaluation is still in the dark. This raises a fundamental question:

**Can the performance of generative models be evaluated statistically from finite samples?**

This question is challenging for several reasons. First, unlike the error rate in supervised learning, the performance metric for generative models is not well specified. Second, even when a metric is specified, it may not be evaluable from finite samples.

For example, one commonly considered metric is cross-entropy, defined as $H(q^\star, q) = \mathbb{E}_{x \sim q^\star}[-\log q(x)]$, where $x$ is a document and $q^\star, q$ are distributions over documents. A commonly used scoring method to evaluate cross-entropy is perplexity, defined as $2^{-\frac{1}{n} \sum_{i=1}^{n} \log q(x_i)}$ for IID documents $x_i \sim q^\star$. However, perplexity can fail to evaluate cross-entropy because $\log q(x)$ is typically unbounded and possibly heavy-tailed: rare events with small $q(x)$ yield arbitrarily large losses, invalidating standard concentration

inequalities. As a result, perplexity can remain highly variable and systematically misrepresent the true cross-entropy, even with large datasets.

Another example comes from statistical learning theory: one of the most fundamental performance metrics is Total Variation Distance (TV), defined as $\mathrm{TV}(q^\star, q) = \frac{1}{2} \int_{\mathcal{X}} |q^\star(x) - q(x)| dx$[1]. TV is a natural metric for evaluating the distance between two distributions. However, as shown in prior work, when the domain is huge, TV is not estimable, which means that given two models, we cannot determine which one has lower TV distance to $q^\star$ even when the test data size is large.

In this work, we introduce a theoretical framework for evaluating generative models and establish evaluability results for a range of commonly used metrics. We study two main categories of performance metrics: test-based metrics and $\alpha$-Rényi divergences (for $\alpha > 1$).

Test-based metrics evaluate models by applying test functions that map each data point to a real value. For example, a correctness test for code returns $1$ if the code is correct, and a length test for an email returns its length. A single test metric compares the difference between the expected test result under the ground truth distribution and the model distribution. Integral probability metrics (IPMs) consider a class of such tests and take the maximum difference over all tests in the class. We show that the evaluability of IPMs is closely tied to measures of statistical complexity of the test class: finite VC dimension for binary tests and finite fat-shattering dimension for real-valued tests are sufficient for strong evaluability.

The class of $\alpha$-Rényi divergences directly measures distributional similarity and can be unbounded. We establish that such metrics are fundamentally not evaluable from finite samples, as their values can be determined by rare events that are undetectable in practice.

**Our Contributions** A majority of results are summarized in Table 1. Our contributions are four-fold.

- **Theoretical Framework for Evaluability.** We mathematically define evaluability as a fundamental property that determines whether a performance metric can be reliably evaluated from finite samples. We introduce two levels of evaluability: a metric is *strongly evaluable* if it can be evaluated with arbitrary precision from finite samples, and *weakly evaluable* if it can be evaluated up to multiplicative and additive approximation errors.

- **Nearly Complete Characterization of IPM Evaluability.** We establish comprehensive evaluability results for integral probability metrics (IPMs), a broad class

---

[1]If $\mathcal{X}$ is finite, TV can instead be written as a sum

of test-based evaluation metrics. Our main findings are: (1) Bounded IPMs are always weakly evaluable, regardless of the complexity of the test class, providing a guarantee that these metrics can at least be evaluated approximately; (2) For binary test classes, we establish a strict dichotomy: IPMs are strongly evaluable if and only if the test class has finite VC dimension, otherwise they are only weakly evaluable with a factor-3 approximation (and cannot be evaluated with a better multiplicative factor) (3) For bounded real-valued test classes, IPMs are strongly evaluable when the test class has finite $\gamma$-fat shattering dimension for all $\gamma > 0$ but it might require arbitrarily large sample size.

- **Negative Results for Rényi divergences** We show that the $\alpha$-Rényi divergences, KL divergence (the special case obtained as $\alpha \to 1$), and coverage profile (Chen et al., 2025) (a truncated version of Rényi divergence designed to address its limitations) are not evaluable from finite samples. This reveals a fundamental limitation: metrics that directly measure distributional similarity cannot be reliably evaluated, as their values can be critically determined by rare events that are undetectable in finite samples.

- **Analysis of Perplexity as a Scoring Method.** While perplexity cannot reliably evaluate cross-entropy (which is equivalent to KL divergence up to a constant), we analyze when perplexity can serve as a valid evaluation method in some scenarios. Specifically, we show that perplexity can evaluate total variation distance under certain conditions, while being unable to evaluate a restricted form of KL divergence. In particular, our results highlight the properties and limitations of this commonly used scoring method.

## 1.1. Related Work

**Uniform Convergence in Supervised Learning** A fundamental question in statistical learning theory centers around when population-level quantities can be reliably estimated from finite samples. Foundational results from Alon et al. (1997); Vapnik (1995); Bartlett et al. (1996) establish uniform convergence guarantees for bounded losses and test statistics through combinatorial measures such as the Vapnik-Chervonenkis (VC) dimension for binary-valued function classes and the $\gamma$-fat-shattering dimension for real-valued classes. These results underpin the idea of statistical evaluation in supervised learning settings, where empirical risk is a proxy for population level risk. However, this line of work is primarily concerned with estimating a single population-level quantity, rather than reliably ranking models using empirical evaluation algorithms. Our framework aims to formally characterize the latter problem.

*Table 1.* Summary of Results

| Evaluation Metric | Evaluability Guarantee |
|---|---|
| IPM w.r.t binary-valued $\mathcal{F}$ | Estimable $\Rightarrow$ strongly evaluable if $\text{VC}(\mathcal{F}) < \infty$; 
 3-weakly evaluable if $\text{VC}(\mathcal{F}) = \infty$; (Theorem 3.2) |
| IPM w.r.t. real-valued $\mathcal{F}$ | Estimable $\Rightarrow$ strongly evaluable if $\text{Pdim}_\gamma(\mathcal{F}) < \infty$ for all $\gamma \in (0, 1/2]$; 
 3-weakly evaluable if $\text{Pdim}_\gamma(\mathcal{F})$ is unbounded for some $\gamma \in (0, 1/2]$ (Proposition 3.4) |
| Fixed Statistical Test Metrics w.r.t test function $g$ | Estimable $\Rightarrow$ strongly evaluable when $g$ is bounded; 
 Not weakly evaluable when $g$ is sufficiently unbounded (Theorem 3.8) |
| $\alpha$-Rényi divergence ($\alpha > 1$) | Not weakly evaluable (Theorem 4.2) |
| Coverage Profile (Chen et al., 2025) | Not weakly evaluable (Proposition 4.5); 
 Strongly evaluable under additional conditions (Theorem 4.8) |
| Total Variation Distance | Not weakly evaluable by perplexity score (Theorem 5.3); 
 Strongly evaluable under additional conditions (Proposition 5.6) |
| $\beta$-Restricted KL Divergence | Not weakly evaluable by perplexity score (Theorem 5.3); 
 Not strongly evaluable (Theorem 5.4) |

**IPMs and Density Estimation** Integral Probability Metrics (IPMs), such as total variation distance and Wasserstein distance, are a well-studied class of distances between distributions (Müller, 1997). They have been analyzed extensively in the context of density estimation, where the IPM is the Total-Variation distance. In particular, Yatracos (1985) shows that there exists an algorithm that, given samples from a ground truth distribution $p$ and class of distributions $Q$, outputs the best distribution in $Q$, up to a multiplicative factor of 3. Bousquet et al. (2019) show that this result is optimal, in the sense that no proper algorithm can achieve a multiplicative approximation factor $c < 3$, even when $Q$ consists of only two distributions. Our work builds on these results by studying IPMs through the lens of evaluability.

**Perplexity and Other Common Score Functions** The perplexity score is one of the most commonly used metrics to evaluate language models on samples, but growing empirical evidence suggests that a good perplexity score may not correlate well with downstream performance (Fang et al., 2025; Hu et al., 2024). Motivated by the instability of raw likelihood-based evaluation, several works have proposed modified perplexity and divergence-based metrics that suppress the influence of rare events. In particular, Pillutla et al. (2021) survey smoothed variants of perplexity, such as $\varepsilon$-perplexity, as sparsity-aware likelihood scores introduced by Martins et al. (2020). Our framework and results provide a theoretical perspective on these issues with the perplexity score and the techniques used to alleviate them.

**Evaluation via Precision–Recall Trade-off** A parallel line of work advocates for a two-dimensional approach to evaluating generative models. Bousquet et al. (2019)

posit that the notions of precision and recall in information retrieval can be used to decouple the evaluation process and capture multiple failure modes. Djolonga et al. (2020) extend this to a general evaluation framework that captures the trade-off between precision and recall.

## 2. Problem Setup

Let $\mathcal{X}$ denote the data space. A model is a probability distribution over $\mathcal{X}$. Let $\mathcal{M} = \Delta(\mathcal{X})$ denote the set of all probability distributions over $\mathcal{X}$, which is our model space. Assume there exists a ground-truth model $q^\star \in \mathcal{M}$.

When evaluating a model $q \in \mathcal{M}$, we compare it to the ground-truth $q^\star$ with respect to an *evaluation metric*.

**Definition 2.1** (Evaluation Metric). Given a ground-truth model $q^\star$ and a model $q$ to be evaluated, an *evaluation metric* $f : \mathcal{M}^2 \mapsto \mathbb{R}_{\geq 0} \cup \{\infty\}$ assigns a non-negative real-valued score $f(q, q^\star)$ to the model $q$.

An evaluation algorithm is given two models $\{q_1, q_2\}$ to compare, together with IID evaluation data sampled from the ground-truth distribution, and outputs the model it believes to be better. A common approach is to evaluate models using a score function.

**Definition 2.2** (Score Function). A *score function* $s : \mathcal{M} \times \mathcal{X}^* \mapsto \mathbb{R} \cup \{\infty\}$, given a model $q$ and a set of evaluation data $S_{\text{eval}}$, outputs a real-valued score $s(q, S_{\text{eval}})$ of this model.

One can induce an evaluation algorithm $\mathcal{A}$ by a score function by outputting the model with the smaller score, breaking ties randomly. Now, we relate evaluation algorithms that operate on finite samples to population-level evaluation metrics via the notion of evaluability. Informally, evalua-

bility means that given enough data from the ground-truth $q^\star$, there exists an evaluation algorithm that can reliably compare any two models in terms of their evaluation metric.

**Definition 2.3** (Evaluability). For any evaluation metric $f : \mathcal{M}^2 \mapsto \mathbb{R}_{\geq 0} \cup \{\infty\}$ and $c \geq 1$, we say $f$ is $c$-weakly evaluable if there exists a function $m_{\text{evl},f} : (0,1)^2 \times \mathbb{R}_{\geq 1} \mapsto \mathbb{N}$ and an evaluation algorithm $\mathcal{A}$ such that for every pair of models $q_1, q_2 \in \mathcal{M}$, every $\varepsilon, \delta \in (0,1)$ and every ground-truth model $q^\star$, given IID evaluation data $S_{\text{eval}} = \{x_1, \ldots, x_m\}$ of size $m \geq m_{\text{evl},f}(\varepsilon, \delta, c)$ with $x_i \sim q^\star$, with probability at least $1 - \delta$,

$$\mathcal{A}(\{q_1, q_2\}, S_{\text{eval}}) = q_1 \implies f(q_1, q^\star) \leq c \cdot f(q_2, q^\star) + \varepsilon \,,$$

and

$$\mathcal{A}(\{q_1, q_2\}, S_{\text{eval}}) = q_2 \implies f(q_2, q^\star) \leq c \cdot f(q_1, q^\star) + \varepsilon \,.$$

We say $f$ is strongly evaluable when $c = 1$. We say $f$ is not weakly evaluable if it is not $c$-weakly evaluable for any $c \geq 1$.

Note that evaluability is closely related with the commonly studied notion of estimability.

**Definition 2.4** (Estimability). For any finite-valued evaluation metric $f : \mathcal{M}^2 \mapsto \mathbb{R}_{\geq 0}$, we say $f$ is estimable if there exists a function $m_{\text{est},f} : (0,1)^2 \mapsto \mathbb{N}$ and a score function $s : \mathcal{M} \times \mathcal{X}^* \mapsto \mathbb{R}$ such that for any $\varepsilon, \delta \in (0,1)$, for any ground-truth model $q^\star$ and any model $q \in \mathcal{M}$, given IID evaluation data $S_{\text{eval}} = \{x_1, \ldots, x_m\}$ of size $m \geq m_{\text{est},f}(\varepsilon, \delta)$ with $x_i \sim q^\star$, with probability at least $1 - \delta$,

$$|s(q, S_{\text{eval}}) - f(q, q^\star)| \leq \varepsilon \,.$$

If $f$ is estimable by some score function $s$, then it is strongly evaluable by an evaluation algorithm induced by the same $s$. Indeed, if a score function $s$ uniformly approximates the value of an evaluation metric $f(\cdot, q^\star)$ to arbitrary accuracy from finite samples, then comparing the estimated values for two models suffices to recover their true ordering up to an additive error, thus recovering strong evaluability.

## 3. Test-Based Evaluation Metrics

We first study test-based evaluation metrics, which are parametrized by statistical tests that aim to distinguish a candidate model from the ground-truth. This perspective reflects how generative models are evaluated in practice: rather than directly comparing models via densities, evaluation proceeds by probing models through a series of tests.

In Section 3.1, we consider a class of bounded test functions and discuss the dependence of evaluability on the complexity of the test function class. In Section 3.2, we consider a single fixed but possibly unbounded test and discuss the dependency of evaluability on the magnitude of the test function.

### 3.1. Integral Probability Metrics

We now instantiate our framework for Integral Probability Metrics (IPMs), helping build intuition about how the complexity of a class of test functions dictates the evaluability of the corresponding IPM.

**Definition 3.1** (IPMs). Given a class $\mathcal{F}$ of $\mathcal{X} \mapsto [0,1]$ functions, the IPM w.r.t. $\mathcal{F}$ is

$$d_{\mathcal{F}}(q^\star, q) = \sup_{\phi \in \mathcal{F}} |\mathbb{E}_{x \sim q^\star}[\phi(x)] - \mathbb{E}_{x \sim q}[\phi(x)]| \,.$$

Each IPM is parametrized by a class of test functions $\mathcal{F}$ and measures the largest statistical discrepancy between two models that can be witnessed by a test function in the class. A model performs well with respect to $d_{\mathcal{F}}(q^\star, \cdot)$ if it behaves similarly to $q^\star$ on all relevant tests. This raises an interesting tradeoff between the richness of the test class $\mathcal{F}$ and the estimability (and evaluability) guarantees about $d_{\mathcal{F}}(q^\star, \cdot)$. A more expressive test class $\mathcal{F}$ allows for more degrees of freedom in terms of distinguishing models. However, the corresponding IPM becomes more difficult to reliably approximate. We formalize this tradeoff for two different complexity measures of $\mathcal{F}$: the VC Dimension and the $\gamma$-fat-shattering dimension.

Our first result considers the regime where tests in $\mathcal{F}$ are binary valued. Note that when $\mathcal{F}$ contains all binary functions, the corresponding IPM is the TV distance. In the binary case, the richness of $\mathcal{F}$ can be characterized by the VC Dimension[2]. The following result establishes a strict dichotomy between estimability (strong evaluability) and 3-weak evaluability characterized entirely by the VC dimension of $\mathcal{F}$. This extends a result from Bousquet et al. (2019), which implies this dichotomy for TV distance.

**Theorem 3.2.** *Consider $\mathcal{F} \subset \{0,1\}^{\mathcal{X}}$ being a class of binary functions. There is a dichotomy of evaluability of $d_{\mathcal{F}}$ for all binary function class $\mathcal{F}$:*

- *If $\mathcal{F}$ has finite VC dimension $\text{VCdim}(\mathcal{F}) < \infty$, $d_{\mathcal{F}}$ is estimable, thus strongly evaluable with sample complexity*

$$O\left(\frac{d \log(1/\varepsilon) + \log(1/\delta)}{\varepsilon^2}\right) \,,$$

*where $d = \text{VCdim}(\mathcal{F})$.*

- *If $\mathcal{F}$ has unbounded VC dimension (i.e., $\mathcal{F}$ shatters arbitrarily large sets), then $d_{\mathcal{F}}$ is 3-weakly evaluable with sample complexity*

$$\min\left\{O\left(\frac{2 + \log(1/\delta)}{\varepsilon^2}\right), \widetilde{O}\left(8\frac{\log^{3/2}(1/\delta)}{\varepsilon^{5/2}}\right)\right\} \,.$$

---

[2]Recall that $\text{VCdim}(\mathcal{F})$ is the largest size of $X \subseteq \mathcal{X}$ s.t. every $c : X \to \{0,1\}$ has an extension in $\mathcal{F}$.

*And there does not exist a $c' < 3$ such that $d_{\mathcal{F}}(q^{\star}, \cdot)$ is $c'$-weakly evaluable.* [3]

When the test class has finite VC dimension, we can obtain a $\varepsilon$-accurate estimate of $\mathbb{E}_{x \sim q^{\star}}[\phi(x)]$ for each $\phi \in \mathcal{F}$ by uniform convergence. In contrast, when the test class has unbounded VC dimension, strong evaluability is impossible, yet a weaker form of evaluability remains achievable. In other words, we can still have a coarse ranking of models even when estimating the IPM is not possible.

Now, extending to bounded real-valued test classes (w.l.o.g mapping from $\mathcal{X}$ to $[0, 1]$), we replace VC dimension with the fat-shattering dimension.

**Definition 3.3** ($\gamma$-Fat-Shattering Dimension). Given a class $\mathcal{F} \subset [0, 1]^{\mathcal{X}}$ of real-valued functions that map from $\mathcal{X}$ to $[0, 1]$, we say that a subset $\{x_1, ..., x_n\} \subseteq \mathcal{X}$ is $\gamma$-shattered by $\mathcal{F}$ if there exist thresholds $r_1, ..., r_n \in [0, 1]$ such that for all labeling vectors $y \in \{\pm 1\}^n$, there exists $g \in \mathcal{F}$ such that

$$y_i = +1 \implies g(x_i) \geq r_i + \gamma \ , \ y_i = -1 \implies g(x_i) \leq r_i - \gamma.$$

The $\gamma$-fat-shattering-dimension of $\mathcal{F}$, denoted as $\text{Pdim}_{\gamma}(\mathcal{F})$, is the size of the largest set that can be $\gamma$-shattered by $\mathcal{F}$.

**Proposition 3.4.** *Given a function class $\mathcal{F} \subset [0, 1]^{\mathcal{X}}$, the IPM $d_{\mathcal{F}}$ satisfies:*

- *If $\mathcal{F}$ has finite $\gamma$-fat-shattering dimension $\text{Pdim}_{\gamma}(\mathcal{F}) < \infty$ for all $\gamma \in (0, \frac{1}{2})$, $d_{\mathcal{F}}$ is estimable, thus strongly evaluable with sample complexity*

$$O\left(\frac{1}{\varepsilon^2}\left(d \log^2 \frac{4d}{\varepsilon^2} + \log \frac{1}{\delta}\right)\right),$$

  *where $d = \text{Pdim}_{\varepsilon/24}(\mathcal{F})$*

- *If there exists $\gamma \in (0, \frac{1}{2})$ such that $\mathcal{F}$ has arbitrarily large $\gamma$-fat-shattering dimension (i.e., for any $n \in \mathbb{N}$, there exists a set $X_n \subseteq \mathcal{X}$ with $|X_n| = n$ such that $\mathcal{F}$ can $\gamma$-fat-shatter $X_n$), $d_{\mathcal{F}}$ is 3-weakly evaluable.*

Note that this result is weaker than Theorem 3.2 since in the second case, we do not establish that $d_{\mathcal{F}}(q^{\star}, q)$ is not $c'$-weakly evaluable for $c' < 3$. We leave this as an open question.

---

[3]The lower bound construction immediately provides a lower bound for the Hellinger Distance as well. In particular, the Hellinger Distance is not $c$-weakly evaluable for any $c \lessapprox 1.1233$. The corresponding upper bound is less clear and would be an interesting direction for future work.

### 3.1.1. THE SAMPLE COMPLEXITY OF ESTIMATING IPMS

In classical supervised learning, once a hypothesis class is learnable, its sample complexity is governed by a small number of canonical rates determined by a single complexity parameter, such as the VC dimension. Specifically, for every learnable class $\mathcal{H}$, the optimal sample complexity to achieve error $\varepsilon$ scales as $\text{VCdim}(\mathcal{H})/\varepsilon$ in the realizable case, and $\text{VCdim}(\mathcal{H})/\varepsilon^2$ in the agnostic case.

For estimable IPMs, and evaluation metrics in general, it is a priori unclear whether an analogous such finite taxonomy of rates exists. One might hope that estimable metrics admit a finite number of sample-complexity behaviors, so that every estimable metric can be evaluated with sample complexity comparable to one of a fixed collection of functions of the desired accuracy. The following definition formalizes this notion.

**Definition 3.5** (Finite taxonomy of sample complexities). We say that a class of evaluation metrics $\mathcal{C}$ admits a *finite taxonomy of sample complexities* if there exists a finite (or countable) list of functions $M_1(\varepsilon, \delta), M_2(\varepsilon, \delta), \ldots$ such that for every estimable metric $f \in \mathcal{C}$, there exists some index $i$ and an estimator with the property that for every pair of distributions $q_1, q_2 \in \mathcal{M}$, the sample complexity of estimating $f(q_1, q_2)$ is at most $M_i(\varepsilon, \delta)$ for all sufficiently small $\varepsilon, \delta > 0$ (where the threshold for "sufficiently small" may depend on $q_1, q_2$).

Informally, each function $M_i$ represents a distinct sample complexity regime. If such a taxonomy existed, then although different metrics might require different constants or estimators, their asymptotic dependence on the accuracy parameter would be drawn from a fixed list. In this sense, the definition captures whether there is a VC-dimension–like parameter that captures the sample complexity of estimating (and thus strongly evaluating) evaluation metrics. The following result shows that in the case of estimable IPMs, no such parameter exists.

**Theorem 3.6.** *There does not exist a finite taxonomy of sample complexities for estimable IPMs. More precisely, for any sequence of functions $M_1(\varepsilon, \delta), M_2(\varepsilon, \delta), \ldots$ (finite or countable), and for any $i \in \mathbb{N}$, there exists an estimable IPM $d_{\mathcal{F}}$ (with test function class $\mathcal{F}$) and distributions $q_1, q_2 \in \mathcal{M}$ such that the estimation sample complexity for $d_{\mathcal{F}}(q_1, q_2)$ is greater than $M_i(\varepsilon, \delta)$ for all sufficiently small $\varepsilon, \delta > 0$.*

### 3.2. Fixed Statistical Tests

The second class of test-based evaluation metrics we consider are fixed statistical tests.

**Definition 3.7** (Fixed Statistical Test). Given any test function $g : \mathcal{X} \mapsto \mathbb{R}$, we define the fixed statistical test w.r.t $g$

as

$$f^g_{\text{fixed}}(q, q^\star) = |\mathbb{E}_{x\sim q^\star}[g(x)] - \mathbb{E}_{x\sim q}[g(x)]|.$$

Here, we admit a single test function as opposed to a larger class, but we do not place any boundedness assumptions on $g$. The evaluability guarantees on $f^g_{\text{fixed}}$ depend on the boundedness of $g$.

**Theorem 3.8.** *Given a test function $g : \mathcal{X} \mapsto \mathbb{R}$,*

- *If there exists a fixed $B < \infty$ such that $|g(x)| \leq B$ for all $x \in \mathcal{X}$, then $f^g_{fixed}$ is estimable, thus strongly evaluable with sample complexity*

$$O\left(\frac{B^2 \log\frac{1}{\delta}}{\varepsilon^2}\right).$$

- *If there exists $x_2 \in \mathcal{X}$ with $0 < |g(x_2)| < \infty$ such that for every $B > 0$, there exists $x_1 \in \mathcal{X}$ with $|g(x_1)| > B|g(x_2)|$, then $f^g_{fixed}$ is not weakly evaluable.*

## 4. Rényi divergences

In this section, we turn to studying evaluation metrics that directly compare distributions. In particular, we focus on the canonical and representative Rényi divergences, introduced in Rényi (1961). These metrics are a unifying family that includes the KL divergence as a special case. Unlike test-based metrics, these divergences compare distributions pointwise using likelihood ratios. Rényi divergences offer a more direct notion of distributional fidelity as opposed to test-based metrics and have important applications in statistics, information theory, and machine learning (Liese & Vajda, 2006; van Erven & Harremoës, 2012; Huang et al., 2024). A central artifact of these divergences that contrasts with test-based metrics is that they can be critically determined by rare points and thus become unbounded. Our results show that this property is fundamentally incompatible with evaluability under our framework, and it cannot be easily fixed by truncation.

### 4.1. Rényi divergences

The main class of metrics we focus on is the $\alpha$-Rényi divergences.

**Definition 4.1** (Rényi divergence). Given $\alpha > 1$, the $\alpha$-Rényi divergence is defined as

$$f_{\alpha-\text{Rényi}}(q, q^\star) = \frac{1}{\alpha - 1} \log \sum_{x \in \mathcal{X}} q^\star(x)^\alpha q(x)^{1-\alpha}.$$

Our main result here is that the $\alpha$-Rényi divergence is not weakly evaluable for any $\alpha > 1$.

**Theorem 4.2.** *For any $\alpha > 1$, $f_{\alpha-\text{Rényi}}$ is not weakly evaluable.*

**Proof Sketch** Consider the following construction of $q_1, q_2$. Given any $\eta \in (0, 1)$ and $M > 0$, define

$$q_1(x) = \begin{cases} 1 - \eta - \eta e^{-M} & x = x_0 \\ \eta e^{-M} & x = x_1 \\ \eta & x = x_2, \end{cases}$$

and

$$q_2(x) = \begin{cases} 1 - \eta - \eta e^{-M} & x = x_0 \\ \eta & x = x_1 \\ \eta e^{-M} & x = x_2. \end{cases}$$

In this construction, $q_1$ and $q_2$ agree on a majority of the mass placed on $x_0$, but are flipped on $x_1, x_2$. The $\alpha$-Rényi divergence $f_{\alpha-\text{Rényi}}(q_1, q_2) = f_{\alpha-\text{Rényi}}(q_2, q_1)$ is large when $M$ is large. When $\eta$ is small enough, $x_1$ and $x_2$ will never be observed in finite samples, no matter whether the ground-truth $q^\star$ is $q_1$ or $q_2$.

Note that KL divergence $f_{\text{KL}}(q, q^\star) := \text{KL}(q^\star \| q)$ is the limit of the $\alpha$-Rényi divergence as $\alpha \to 1$. The same construction in the proof of Theorem 4.2 gives us the following corollary.

**Corollary 4.3.** $f_{\text{KL}}$ *is not weakly evaluable.*

These two divergences share a common pathology that our construction exploits — their value can become unbounded if a model places negligible mass compared to the ground truth on even a single point. Additionally, these metrics are inherently one-sided, whereas the test-based metrics we discussed are symmetric. More specifically, the value of the $\alpha$-Rényi divergence or KL divergence can only become unbounded through points where $q^\star$ is nonzero but $q$ is vanishingly small. In this sense, these divergences penalize undercoverage of the ground-truth distribution rather than generic mismatch. We believe that our negative results for the $\alpha$-Rényi and KL divergences extend to a broader subclass of tail-sensitive $f$-divergences. However, we know that this does not hold in general for all $f$-divergences, as Theorem 3.2 shows that the TV distance is 3-weakly evaluable. It would be interesting to explore the characterization of evaluability for general $f$-divergences in future work.

### 4.2. The Coverage Profile

A direct fix of the downfalls of the $\alpha$-Rényi divergence is to "truncate" the metric in some way. This is motivated by the fact that evaluation should focus on the bulk of the data rather than being destabilized by rare events. Here we show that this simple fix does not immediately lead to evaluability in general, as evidenced by the following metric: the coverage profile, introduced in Chen et al. (2025).

**Definition 4.4** (Coverage Profile). Given a natural number

$N \geq 1$, the coverage profile w.r.t. $N$ is defined as

$$f_{N-\text{Cov}}(q, q^\star) = \mathbb{P}_{x \sim q^\star}\left[\frac{q^\star(x)}{q(x)} \geq N\right].$$

The coverage profile captures a model's ability to sufficiently *cover* rare data points under the ground truth model. However, unlike the $\alpha$-Rényi and KL divergences, it penalizes models uniformly rather than exponentially. Concretely, for a fixed point $x$ and two models $q$ and $q'$ such that $N \leq \frac{q^\star(x)}{q(x)} \leq N + \eta$ for small $\eta > 0$ and $\frac{q^\star(x)}{q'(x)} \gg N$, $q$ and $q'$ are penalized equally for this particular point under the coverage profile even though $q'$ places far less mass on $x$. This is an overcorrection to the issue that appears with the earlier metrics, and we show that $f_{N-\text{Cov}}$ is not evaluable.

**Proposition 4.5.** *For any $N \geq 2$, $f_{N-\text{Cov}}$ is not weakly evaluable.*

**Proof Construction** For some $\gamma > 0$ and $0 < \eta \ll \gamma$, define candidate models

$$q_1(x_0) = \frac{1 - \gamma}{N}, \qquad q_1(x_1) = 1 - \frac{1 - \gamma}{N},$$

and

$$q_2(x_0) = 1 - \frac{\gamma}{N}, \qquad q_2(x_1) = \frac{\gamma}{N}.$$

Define two possible ground truths

$$q_3(x_0) = 1 - \gamma, \qquad q_3(x_1) = \gamma,$$

and

$$q_4(x_0) = 1 - \gamma - \eta, \qquad q_4(x_1) = \gamma + \eta.$$

Under $q_3$, $q_2$ is much better than $q_1$ in coverage profile, whereas under $q_4$, $q_1$ is better than $q_2$. However, $q_3$ and $q_4$ have total variation distance $\eta$, and hence their $m$-sample laws are indistinguishable when $\eta \ll 1/m$. Therefore no evaluation algorithm can reliably choose the correct model in both worlds. Note that under our proof construction, the negative result holds even under the following lower-bound (finite support) assumption on $q^\star$.

**Assumption 4.6.** There exists a $\gamma > 0$ such that $q^\star(x) \geq \gamma$ for all $x \in \text{supp}(q^\star)$.

At a high level, the coverage profile is not evaluable even under Assumption 4.6 because it is discontinuous around the threshold $N$. More specifically, it does not penalize models that come arbitrarily close to but remain below the threshold and penalizes models that exceed the threshold equally. Hence, models that are indistinguishable with finite samples could actually have very different coverage profiles.

This motivates the need for a margin assumption that prevents the ratio $\frac{q^\star(x)}{q(x)}$ from clustering too close to the threshold $N$. We present such an assumption along with the corresponding positive result about the evaluability of the coverage profile below.

**Assumption 4.7.** We say $(q^\star, q)$ have an $(N, \alpha)$ margin if $\left|\frac{q^\star(x)}{q(x)} - N\right| \geq \alpha$ for all $x \in \text{supp}(q^\star)$.

**Theorem 4.8.** *Fix $N \geq 2$, $\varepsilon, \delta \in (0, 1)$, and any two candidate models $q_1, q_2 \in \mathcal{M}$ such that $(q^\star, q)$ have an $(N, \alpha)$ margin for some fixed $\alpha > 0$ for each $q \in \{q_1, q_2\}$. Then, there exists an evaluation algorithm $\mathcal{A}$ such that if $q^\star$ satisfies Assumption 4.6 for some fixed $\gamma > 0$, given evaluation data $S_{eval} = \{x_1, \ldots, x_m\}$ of size*

$$m \geq \max\left(\frac{3(N + \alpha)^2}{\alpha^2 \gamma} \log \frac{4}{\gamma \delta}, \quad \frac{2}{\varepsilon^2} \log \frac{8}{\delta}\right),$$

*with $x_i \sim q^\star$, with probability at least $1 - \delta$,*

$$\mathcal{A}(\{q_1, q_2\}, S_{eval}) = q_1$$
$$\implies f_{N-\text{Cov}}(q_1, q^\star) \leq f_{N-\text{Cov}}(q_2, q^\star) + \varepsilon$$

*and*

$$\mathcal{A}(\{q_1, q_2\}, S_{eval}) = q_2$$
$$\implies f_{N-\text{Cov}}(q_2, q^\star) \leq f_{N-\text{Cov}}(q_1, q^\star) + \varepsilon$$

Note that the final statement in Theorem 4.8 is identical to the definition of strong evaluability. The difference is that rather than the result holding for all possible ground truths and pairs of candidate models, it only holds in settings where Assumption 4.6 and Assumption 4.7 are satisfied.

# 5. Perplexity as a Score Function

In this section, we highlight the limitations of the perplexity score. We have already shown that KL divergence is not weakly evaluable, thus not evaluable by perplexity. Then we focus on other potential metrics, specifically on TV distance and a restricted notion of KL divergence. Through our results, we provide intuition for why perplexity might not be practical as a score function in general. In the following, we present our results in terms of the negative log-likelihood (nll) score rather than perplexity; since perplexity is simply the exponential of the nll, both scores induce the same ordering over models and are therefore equivalent for the purposes of evaluability. We formally define the nll score as

$$\text{nll}(q, S_{\text{eval}}) = -\frac{1}{m} \sum_{i=1}^{m} \log q(x_i).$$

**TV Distance** We define Total Variation distance as an evaluation metric as $f_{\text{TV}}(q, q^\star) := \text{TV}(q^\star, q)$. $f_{\text{TV}}$ is a symmetric and bounded metric and equivalent to IPM w.r.t. $\mathcal{F} = [0, 1]^{\mathcal{X}}$ being all bounded real-valued functions.

**$\beta$-Restricted KL Divergence** From Corollary 4.3, we know that $f_{\text{KL}}$ is not weakly evaluable by any score function,

including the nll score, due to rare events. To eliminate dependence on rare events, we can consider a restricted notion of KL divergence by ignoring a small fraction of samples, potentially resolving the unboundedness issue. We define this below.

**Definition 5.1** (Restricted KL Divergences). Given a subset $E \subseteq \mathcal{X}$, define the KL divergence restricted to $E$ as

$$\text{KL}_E(q^\star, q) = \left[ \mathbb{E}_{x \sim q^\star(\cdot|E)} \left[ \log \frac{q^\star(x)}{q(x)} \right] \right]_+ .$$

Now, we define a class of evaluation metrics parametrized by the amount of mass that we remove from $\mathcal{X}$ under $q^\star$.

**Definition 5.2.** Given $\beta \in \left(0, \frac{1}{2}\right)$, the $\beta$-Restricted KL divergence is defined as

$$f_{\beta-\text{KL}}(q, q^\star) := \inf_{E \subseteq \mathcal{X}: q^\star(E) \geq 1-\beta} \text{KL}_E(q^\star, q).$$

Intuitively, $f_{\beta-\text{KL}}$ measures the smallest positive average log-likelihood ratio on a subset carrying at least $1 - \beta$ mass under $q^\star$. This is an unnormalized restriction of the KL integrand rather than explicitly being a KL divergence between the conditional distributions.

## 5.1. Evaluability Results of Perplexity

Our first result reveals that without additional assumptions, the nll score fails to weakly evaluate both $f_{TV}$ and $f_{\beta-\text{KL}}$.

**Theorem 5.3.** $f_{TV}$ *is not weakly evaluable by the nll score. Additionally,* $f_{\beta-\text{KL}}$ *is not weakly evaluable by nll score for any* $\beta \in \left(0, \frac{1}{2}\right)$.

At a high level, both of these metrics are geometrically misaligned from the nll score. In the case of the TV distance, the nll score can be unbounded, while total variation is bounded in $[0, 1]$. Further, the nll score is fundamentally one-sided in the sense that it severely penalizes models for undercounting mass relative to $q^\star$, whereas TV distance is entirely governed by how much mass is "moved" by $q$ with respect to $q^\star$, regardless of the direction. Our proof construction capitalizes on this mismatch, causing the nll score to misrank two models with a constant gap in their TV distance to $q^\star$.

While $f_{\beta-\text{KL}}(\cdot, q^\star)$ can still be unbounded unlike TV distance, the nll score is still highly sensitive to single points while $f_{\beta-\text{KL}}$ allows us to ignore up to $\beta$ mass of points. This artifact of $f_{\beta-\text{KL}}$ perhaps makes it more conducive to being evaluable in general compared to standard KL divergence, but this is not exploited by the nll score. However, the following result shows that $f_{\beta-\text{KL}}$ is not strongly evaluable in general, so any positive result would be a weak one.

**Theorem 5.4.** $f_{\beta-\text{KL}}$ *is not strongly evaluable.*

Armed with this intuition, we investigate a natural follow-up question: *Under what conditions does the nll score reliably evaluate either of these metrics?* Perhaps, if one of our candidate models is close to $q^\star$ in a certain sense, we can ensure that the nll ratio $\frac{q^\star(x)}{q(x)}$ is bounded.

The failure of the nll score to evaluate TV distance stems from extreme likelihood distortions on rare events that do not impact the TV calculation in the same way. To preclude this pathology, we impose the following bounded ratio assumption.

**Assumption 5.5** ($\Delta$-Closeness in Ratio). $(q^\star, q)$ are $\Delta$-close in ratio if there exists fixed $\Delta \in (0, \frac{1}{2})$ such that

$$(1 - \Delta)q^\star(x) \leq q(x) \leq (1 + \Delta)q^\star(x).$$

Note that the above assumption ensures that $q$ cannot be supported on any points that $q^\star$ does not place mass on. This should alleviate the geometric mismatch between the nll score and TV distance because:

- If $q$ "moves" mass onto points that are impossible under $q^\star$, this is included in the calculation for $f_{TV}(q, q^\star)$.

- These points are undetectable by the nll score because they can never be observed in $S_{\text{eval}}$.

Indeed, we are now able to recover a positive evaluability result for $f_{TV}(\cdot, q^\star)$ by nll score.

**Proposition 5.6.** *Fix* $\varepsilon, \delta \in (0, 1)$ *and two candidate models* $q_1, q_2 \in \mathcal{M}$ *such that* $(q^\star, q)$ *are* $\Delta$-*close in ratio for some* $q \in \{q_1, q_2\}$ *and* $\Delta \in (0, \frac{1}{2})$. *Then, if* $\Delta \leq O(\varepsilon^2)$, *there exists an evaluation algorithm* $\mathcal{A}$ *such that given evaluation data* $S_{eval} = \{x_1, \ldots, x_m\}$ *of size*

$$m \geq O\left(\frac{1}{\varepsilon^2} \log \frac{1}{\delta}\right),$$

*with* $x_i \sim q^\star$, *with probability at least* $1 - \delta$,

$$\mathcal{A}(\{q_1, q_2\}, S_{eval}) = q_1 \implies f_{TV}(q_1, q^\star) \leq f_{TV}(q_2, q^\star) + \varepsilon.$$

*and*

$$\mathcal{A}(\{q_1, q_2\}, S_{eval}) = q_2 \implies f_{TV}(q_2, q^\star) \leq f_{TV}(q_1, q^\star) + \varepsilon.$$

Fundamentally, Assumption 5.5 transforms our setting into a semi-realizable one and ensures that one of the models must achieve a bounded nll score close to that of $q^\star$.

Now, it is natural to ask whether we can obtain a similar positive result for $\beta$-restricted KL via additional assumptions. We do not know of any such non-degenerate assumption (beyond assuming that one of the candidate models is $q^\star$ itself). In designing a potential assumption to recover evaluability

for $f_{\beta-\text{KL}}(\cdot, q^\star)$ by the nll score, we contrast it with TV distance. First, a condition in the style of Assumption 5.5 is too strong for this setting since the restricted KL divergence still retains the one-sided property of the standard KL divergence. Additionally, it allows for arbitrary behavior of models on $\beta$ fraction of the mass under $q^\star$. Such an assumption is almost redundant because it rules out behaviors that this metric is designed to tolerate. Instead, when thinking about the notion of a "good" candidate model in this setting, we need a model whose undercoverage is confined to a fixed region of mass under $q^\star$, and on the remaining region, the likelihood ratio is bounded. However, this is still not sufficient, as $f_{\beta-\text{KL}}$ is inherently discontinuous with respect to $\beta$. The metric allows us to ignore arbitrary subsets of mass, and as $\beta$ changes, these subsets can look very different. Neither TV distance nor nll score have this property.

**Discussion** The perplexity (nll) score aggregates per-sample penalties of the form $-\log q(x)$. The score is highly sensitive to the worst case. A single data point that is assigned extremely small probability by $q$ dominates the score, regardless of the model's behavior on the rest of the distribution. This property of the score is fundamentally misaligned with many distributional metrics, such as TV distance and restricted KL divergence, which are robust to small regions of mass under $q^\star$. Overall, our results from this section suggest that the perplexity score is poorly suited as a score function that ranks models by overall distributional similarity.

The perplexity score is able to more reliably evaluate these metrics under assumptions about the existence of a "good" model with respect to the ground-truth, validating its use in practice. However, our results highlight that perplexity cannot be used blindly as an evaluation method without a principled understanding of the underlying candidate models that it is used to rank.

More broadly, the pitfalls of the perplexity score uncover an important principle about evaluability: score functions need to be aligned in some sense with the geometric properties of evaluation metrics. Metrics that tolerate rare errors cannot be evaluated by scores that amplify them. Note that this alignment may still not be sufficient for evaluability, as in the case of similarity-based metrics like the $\alpha$-Rényi divergence and coverage profile.

## 6. Open Questions

**The Relationship Between Estimability and Strong Evaluability** It is easy to see that when an evaluation metric $f$ is estimable, then it is strongly evaluable. In fact, the same score function that estimates $f$ also strongly evaluates $f$. It is unclear whether the converse holds true. That is, does strong evaluability imply estimability, and does

such a statement hold in general or for specific classes of evaluation metrics?

**Strict Dichotomy of Evaluability for Real-Valued IPMs** In Proposition 3.4, we establish a weak dichotomy for IPMs with real-valued, bounded test classes. In particular, we leverage the $\gamma$-fat-shattering dimension as a tool to establish this dichotomy. We prove that even if the $\gamma$-fat-shattering dimension of the test class is unbounded for some $\gamma$, the corresponding IPM is still 3-weakly evaluable. However, we were not able to prove that the factor of 3 is optimal, which would mirror our result in the case where the test class is binary-valued. We believe that this same dichotomy between strong evaluability and 3-weak evaluability exists in the real-valued case, but a tool other than the $\gamma$-fat-shattering dimension might be required.

**Evaluability of $\beta$-Restricted KL Divergence** In discussing the limitations of perplexity as an evaluation method, we introduced the $\beta$-restricted KL divergence and showed that the nll score cannot evaluate this metric and may not be able to under any reasonable conditions. However, this metric is a robust alternative to the standard KL divergence. Any general evaluability guarantees for the $\beta$-restricted KL beyond the nll score are left as an open question.

## Acknowledgments

This project has received funding from the European Research Council (ERC) under the European Union's Horizon 2020 research and innovation program (grant agreement No. 882396), by the Israel Science Foundation, the Yandex Initiative for Machine Learning at Tel Aviv University and a grant from the Tel Aviv University Center for AI and Data Science (TAD).

Shay Moran is a Robert J. Shillman Fellow; he acknowledges support by ISF grant 1225/20, by BSF grant 2018385, by Israel PBC-VATAT, by the Technion Center for Machine Learning and Intelligent Systems (MLIS), and by the European Union (ERC, GENERALIZATION, 101039692). Views and opinions expressed are, however, those of the author(s) only and do not necessarily reflect those of the European Union or the European Research Council Executive Agency. Neither the European Union nor the granting authority can be held responsible for them.

Han Shao acknowledges support from an Adobe Research gift. HS thanks Shanshan Wu and Freda Shi for their thoughtful discussions from an application perspective.

## Impact Statement

This paper presents work whose goal is to advance the field of Machine Learning. There are many potential societal consequences of our work, none which we feel must be specifically highlighted here.

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

## A. Proofs From Section 3

**Theorem 3.2.** *Consider $\mathcal{F} \subset \{0, 1\}^{\mathcal{X}}$ being a class of binary functions. There is a dichotomy of evaluability of $d_{\mathcal{F}}$ for all binary function class $\mathcal{F}$:*

- *If $\mathcal{F}$ has finite VC dimension $\mathrm{VCdim}(\mathcal{F}) < \infty$, $d_{\mathcal{F}}$ is estimable, thus strongly evaluable with sample complexity*

$$O\left(\frac{d\log(1/\varepsilon) + \log(1/\delta)}{\varepsilon^2}\right),$$

  *where $d = \mathrm{VCdim}(\mathcal{F})$.*

- *If $\mathcal{F}$ has unbounded VC dimension (i.e., $\mathcal{F}$ shatters arbitrarily large sets), then $d_{\mathcal{F}}$ is 3-weakly evaluable with sample complexity*

$$\min\left\{O\left(\frac{2 + \log(1/\delta)}{\varepsilon^2}\right), \widetilde{O}\left(8\frac{\log^{3/2}(1/\delta)}{\varepsilon^{5/2}}\right)\right\}.$$

  *And there does not exist a $c' < 3$ such that $d_{\mathcal{F}}(q^{\star}, \cdot)$ is $c'$-weakly evaluable.* [4]

*Proof.* The first item can be proved through standard concentration inequalities and Sauer's Lemma.

For the second item, fix $\varepsilon, \delta \in (0, 1)$ and two models $q_1, q_2 \in \mathcal{M}$. Let $\widehat{q}$ be the output of any algorithm by Bousquet et al. (2019). From Theorem 8 of Bousquet et al. (2019), we know that $\widehat{q}$ satisfies $d_{\mathcal{F}}(\widehat{q}, q) \leq d_{\mathcal{F}}(q^{\star}, q) + \frac{\varepsilon}{2}$ for $q \in \{q_1, q_2\}$, and $d_{\mathcal{F}}(\widehat{q}, q^{\star}) \leq 2\min_{q' \in \{q_1, q_2\}} d_{\mathcal{F}}(q', q^{\star}) + \frac{\varepsilon}{2}$ with probability at least $1 - \delta$ if our sample size

$$m \geq \min\left\{O\left(\frac{2 + \log(1/\delta)}{\varepsilon^2}\right), \widetilde{O}\left(8\frac{\log^{3/2}(1/\delta)}{\varepsilon^{5/2}}\right)\right\}.$$

Fix a dataset $S_{\mathrm{eval}}$ with $m$ samples. Consider the score function $s(q, S_{\mathrm{eval}}) = d_{\mathcal{F}}(\widehat{q}, q)$.

Then, if $s(q_1, S_{\mathrm{eval}}) \leq s(q_2, S_{\mathrm{eval}})$, we have

$$\begin{aligned}
d_{\mathcal{F}}(q^{\star}, q_1) &\leq d_{\mathcal{F}}(\widehat{q}, q_1) + d_{\mathcal{F}}(\widehat{q}, q^{\star}) \\
&\leq d_{\mathcal{F}}(\widehat{q}, q_1) + 2\min_{q' \in \{q_1, q_2\}} d_{\mathcal{F}}(q^{\star}, q') + \frac{\varepsilon}{2} \\
&\leq d_{\mathcal{F}}(\widehat{q}, q_2) + 2\min_{q' \in \{q_1, q_2\}} d_{\mathcal{F}}(q^{\star}, q') + \frac{\varepsilon}{2} \\
&\leq 3\, d_{\mathcal{F}}(q^{\star}, q_2) + \varepsilon\,.
\end{aligned}$$

Similarly, if $s(q_2, S_{\mathrm{eval}}) \leq s(q_1, S_{\mathrm{eval}})$, we have that $d_{\mathcal{F}}(q^{\star}, q_2) \leq 3d_{\mathcal{F}}(q^{\star}, q_1) + \varepsilon$. Hence, we can 3-weakly evaluate $d_{\mathcal{F}}$ using the evaluation algorithm $\mathcal{A}$ induced by $s$.

If there exists an evaluation algorithm $\mathcal{A}$ that can $c'$-weakly evaluate $d_{\mathcal{F}}(q^{\star}, \cdot)$ with $c' < 3$, it implies that we can properly perform density estimation with multiplicative factor of $c' < 3$, which conflicts with the lower bound result in Theorem 19 in Bousquet et al. (2019), in which $d_{\mathcal{F}} = \mathrm{TV}$. □

**Proposition 3.4.** *Given a function class $\mathcal{F} \subset [0, 1]^{\mathcal{X}}$, the IPM $d_{\mathcal{F}}$ satisfies:*

- *If $\mathcal{F}$ has finite $\gamma$-fat-shattering dimension $\mathrm{Pdim}_{\gamma}(\mathcal{F}) < \infty$ for all $\gamma \in (0, \frac{1}{2})$, $d_{\mathcal{F}}$ is estimable, thus strongly evaluable with sample complexity*

$$O\left(\frac{1}{\varepsilon^2}\left(d\log^2\frac{4d}{\varepsilon^2} + \log\frac{1}{\delta}\right)\right),$$

  *where $d = \mathrm{Pdim}_{\varepsilon/24}(\mathcal{F})$*

---

[4] The lower bound construction immediately provides a lower bound for the Hellinger Distance as well. In particular, the Hellinger Distance is not $c$-weakly evaluable for any $c \lessapprox 1.1233$. The corresponding upper bound is less clear and would be an interesting direction for future work.

- *If there exists $\gamma \in (0, \frac{1}{2})$ such that $\mathcal{F}$ has arbitrarily large $\gamma$-fat-shattering dimension (i.e., for any $n \in \mathbb{N}$, there exists a set $X_n \subseteq \mathcal{X}$ with $|X_n| = n$ such that $\mathcal{F}$ can $\gamma$-fat-shatter $X_n$), $d_{\mathcal{F}}$ is 3-weakly evaluable.*

*Proof.* Fix $\varepsilon, \delta \in (0, 1)$. Given $S_{\text{eval}} = \{x_1, ..., x_m\}$, define the empirical distribution $\widehat{q}(x) = \frac{1}{m} \sum_{i=1}^{m} \mathbb{1}[x_i = x]$. From Alon et al. (1997), we have that if $\text{Pdim}_\gamma(\mathcal{F})$ is finite for all $\gamma$, then with probability at least $1 - \delta$, if $m \geq O(\frac{1}{\varepsilon^2} (d \log^2 \frac{4d}{\varepsilon^2} + \log \frac{1}{\delta}))$, where $d = \text{Pdim}_{\varepsilon/24}(\mathcal{F})$,

$$d_{\mathcal{F}}(q^\star, \widehat{q}) \leq \frac{\varepsilon}{2}.$$

Then, consider the score function

$$s(q, S_{\text{eval}}) = d_{\mathcal{F}}(q, \widehat{q}).$$

Now, we have via the reverse triangle inequality that

$$|s(q, S_{\text{eval}}) - d_{\mathcal{F}}(q^\star, q)| = |d_{\mathcal{F}}(q, \widehat{q}) - d_{\mathcal{F}}(q^\star, q)|$$
$$\leq d_{\mathcal{F}}(q^\star, \widehat{q}) \leq \frac{\varepsilon}{2}.$$

Thus, $d_{\mathcal{F}}(q^\star, \cdot)$ is estimable.

To see that this implies strong evaluability, consider any two models $q_1, q_2 \in \mathcal{M}$. If $s(q_1, S_{\text{eval}}) \leq s(q_2, S_{\text{eval}})$, then

$$d_{\mathcal{F}}(q^\star, q_1) \leq d_{\mathcal{F}}(\widehat{q}, q_1) + \frac{\varepsilon}{2} \leq d_{\mathcal{F}}(\widehat{q}, q_2) + \frac{\varepsilon}{2} \leq d_{\mathcal{F}}(q^\star, q_2) + \varepsilon.$$

Similarly, if $s(q_2, S_{\text{eval}}) \leq s(q_1, S_{\text{eval}})$, then $d_{\mathcal{F}}(q^\star, q_2) \leq d_{\mathcal{F}}(q^\star, q_1) + \varepsilon$. The evaluation algorithm $\mathcal{A}$ induced by $s$ strongly evaluates $d_{\mathcal{F}}$.

The proof of the second bullet is identical to the proof of the first part of the second bullet in Theorem 3.2, as all we require is that functions in $\mathcal{F}$ are bounded in $[0, 1]$. $\square$

**Theorem 3.6.** *There does not exist a finite taxonomy of sample complexities for estimable IPMs. More precisely, for any sequence of functions $M_1(\varepsilon, \delta), M_2(\varepsilon, \delta), \ldots$ (finite or countable), and for any $i \in \mathbb{N}$, there exists an estimable IPM $d_{\mathcal{F}}$ (with test function class $\mathcal{F}$) and distributions $q_1, q_2 \in \mathcal{M}$ such that the estimation sample complexity for $d_{\mathcal{F}}(q_1, q_2)$ is greater than $M_i(\varepsilon, \delta)$ for all sufficiently small $\varepsilon, \delta > 0$.*

*Proof.* We consider a domain $\mathcal{X} = \bigsqcup_{k \geq 1} X_k$ partitioned into a disjoint union of finite-size subsets, where each $X_k$ is of size $N_k$ to be determined later. Let $\mathcal{H}_k$ denote the set of all functions mapping from $X_k$ to $\{0, 1\}$. Finally, define $\mathcal{F} := \bigcup_{k \geq 1} \{f_{k,h}(x) = \frac{1}{k} \mathbb{1}\{x \in X_k\} h(x) | h \in \mathcal{H}_k\}$.

We first prove that for any fixed $\gamma > 0$, $\text{Pdim}_\gamma(\mathcal{F}) < \infty$. This implies by Proposition 3.4 that the IPM metric w.r.t $\mathcal{F}$, $d_{\mathcal{F}}$, is estimable. Fix $\gamma > 0$. Now observe that for any $k > \frac{1}{2\gamma}$ and for any $x \in X_k$ and $f \in \mathcal{F}$, $f(x) \in \{0, \frac{1}{k}\}$ and $\frac{1}{k} < 2\gamma$. It follows that $x$ cannot be $\gamma$-fat-shattered by $\mathcal{F}$. Thus, all of the shattered points must come from the union over the subsets $X_k$ for which $k \leq (1/2\gamma)$. Since there are finitely many such $k$'s, the $\gamma$-fat-shattering dimension of $\text{Pdim}_\gamma(\mathcal{F})$ must be finite.

Fix any $k \geq 1$. Observe that restricted to $X_k$, $\mathcal{F}$ is isomorphic to the class $\{\frac{1}{k} h | h \in \mathcal{H}_k\}$, which is the class of all Boolean functions on $N_k$ points, scaled by $\frac{1}{k}$. In this case, the VC dimension of $\mathcal{F}$ restricted to $X_k$ is $N_k$. Now, we invoke the standard sample complexity lower bound for estimating the IPM w.r.t a binary class up to accuracy parameter $\varepsilon$ (Shalev-Shwartz & Ben-David, 2014). Because we are scaling by $\frac{1}{k}$, we require $m \geq c \frac{N_k + \log(1/\delta)}{(k\varepsilon)^2}$, for some universal constant $c > 0$. Now, for each $k \in \mathbb{N}$, consider the accuracy parameter $\varepsilon_k = \frac{1}{k}$ and error parameter $\delta_k = \frac{1}{k}$. Plugging this into our lower bound gives that we need at least $c(N_k + \log k)$ samples. Now, consider any finite or countable sequence of functions $M_1(\varepsilon, \delta), M_2(\varepsilon, \delta), ...$

Choose $N_k = \lceil \frac{1}{c} \max_{1 \leq j \leq k} M_j(\varepsilon_k, \delta_k) \rceil + 1$. We conclude the proof by assuming that a finite taxonomy of sample complexities exists and establishing a contradiction. In particular, there is some function $M_i$ in the sequence such that the sample complexity of estimating $d_{\mathcal{F}}$ up to accuracy $\varepsilon$ is upper bounded by $M_i(\varepsilon, \delta)$ for all sufficiently small $\varepsilon, \delta$. Consider

the sequence of accuracies, $\varepsilon_k, \delta_k$ for all $k \geq i$. At any such $\varepsilon_k, \delta_k$, we know that the sample complexity of estimation is at least $c(N_k + \log k) > M_i(\varepsilon_k, \delta_k)$. Since $\varepsilon_k, \delta_k \to 0$, there are infinitely many arbitrarily small accuracy and error parameters for which the sample complexity of estimation exceeds $M_i(\varepsilon, \delta)$, giving our contradiction. □

**Theorem 3.8.** *Given a test function* $g : \mathcal{X} \mapsto \mathbb{R}$,

- *If there exists a fixed* $B < \infty$ *such that* $|g(x)| \leq B$ *for all* $x \in \mathcal{X}$*, then* $f_{fixed}^g$ *is estimable, thus strongly evaluable with sample complexity*

$$O\left(\frac{B^2 \log \frac{1}{\delta}}{\varepsilon^2}\right).$$

- *If there exists* $x_2 \in \mathcal{X}$ *with* $0 < |g(x_2)| < \infty$ *such that for every* $B > 0$*, there exists* $x_1 \in \mathcal{X}$ *with* $|g(x_1)| > B|g(x_2)|$*, then* $f_{fixed}^g$ *is not weakly evaluable.*

*Proof.* Fix $\varepsilon, \delta \in (0, 1)$, any model $q \in \mathcal{M}$, and a test function $g$ such that there exists $B < \infty$ with $|g(x)| \leq B$ for all $x \in \mathcal{X}$. Consider $S_{\text{eval}} = \{x_1, ..., x_m\}$ sampled from $q^\star$. Define the empirical average

$$\widehat{q} = \frac{1}{m} \sum_{i=1}^m g(x_i).$$

Consider the score function $s(q, S_{\text{eval}}) = |\widehat{q} - \mathbb{E}_{x \sim q}[g(x)]|$. Via a reverse triangle inequality, we have that $|s(q, S_{\text{eval}}) - f_{\text{fixed}}^g| \leq |\widehat{q} - \mathbb{E}_{x \sim q^\star}[g(x)]|$. Now, applying a Hoeffding Bound gives that if $m \geq O\left(\frac{B^2 \log \frac{1}{\delta}}{\varepsilon^2}\right)$, with probability at least $1 - \delta$, $|\widehat{q} - \mathbb{E}_{x \sim q^\star}[g(x)]| < \varepsilon$. It follows that $f_{\text{fixed}}^g(\cdot, q^\star)$ is estimable and thus strongly evaluable. This concludes the proof of the first bullet.

For the second bullet, fix any $c \geq 1$. Suppose for contradiction that $f_{\text{fixed}}^g$ is $c$-weakly evaluable. Let $\mathcal{A}$ be the corresponding evaluation algorithm. Fix $\delta = \frac{1}{4}$ and $\varepsilon = \frac{1}{2}$. Let

$$m = m_{\text{evl}, f_{\text{fixed}}^g}(\varepsilon, \delta, c).$$

Choose $B > 1$ large enough so that

$$\frac{m}{\sqrt{B}} < \frac{1}{4}$$

and

$$\frac{B-1}{\sqrt{B}}|g(x_2)| > \varepsilon.$$

By assumption, there exist $x_1, x_2 \in \mathcal{X}$ such that $|g(x_2)|$ is nonzero and finite and $|g(x_1)| > B|g(x_2)|$.

Consider the following construction:

$$q_1(x) = \begin{cases} \frac{1}{\sqrt{B}} & x = x_1 \\ 1 - \frac{1}{\sqrt{B}} & x = x_2 \end{cases}$$

and

$$q_2(x) = \begin{cases} 0 & x = x_1 \\ 1 & x = x_2 \end{cases}.$$

Now, observe that

$$\text{TV}(q_1, q_2) = \frac{1}{\sqrt{B}}.$$

Thus,

$$\text{TV}(q_1^m, q_2^m) \leq m\text{TV}(q_1, q_2) = \frac{m}{\sqrt{B}} < \frac{1}{4}.$$

Simultaneously, by symmetry of the metric,

$$f_{\text{fixed}}^g(q_1, q_2) = f_{\text{fixed}}^g(q_2, q_1) = \frac{1}{\sqrt{B}}|g(x_1) - g(x_2)| \geq \frac{B-1}{\sqrt{B}}|g(x_2)| > \varepsilon.$$

Under ground truth $q_1$, outputting $q_2$ violates the evaluability definition, since

$$f_{\text{fixed}}^g(q_2, q_1) > \varepsilon = c f_{\text{fixed}}^g(q_1, q_1) + \varepsilon.$$

Therefore,

$$\mathbb{P}_{S_{\text{eval}} \sim q_1^m} \left[ \mathcal{A}(\{q_1, q_2\}, S_{\text{eval}}) = q_1 \right] \geq 1 - \delta.$$

Similarly, under ground truth $q_2$

$$f_{\text{fixed}}^g(q_1, q_2) > \varepsilon = c f_{\text{fixed}}^g(q_2, q_2) + \varepsilon.$$

Therefore,

$$\mathbb{P}_{S_{\text{eval}} \sim q_2^m} \left[ \mathcal{A}(\{q_1, q_2\}, S_{\text{eval}}) = q_1 \right] \leq \delta.$$

Let

$$E = \{ S_{\text{eval}} : \mathcal{A}(\{q_1, q_2\}, S_{\text{eval}}) = q_1 \}.$$

Combining the two displays above gives

$$q_1^m(E) - q_2^m(E) \geq 1 - 2\delta = \frac{1}{2}.$$

Thus,

$$\text{TV}(q_1^m, q_2^m) \geq \frac{1}{2},$$

which contradicts the earlier bound that $\text{TV}(q_1^m, q_2^m) < \frac{1}{4}$. Hence $f_{\text{fixed}}^g$ is not $c$-weakly evaluable. The result follows by the fact that $c$ was chosen arbitrarily. $\square$

## B. Proofs From Section 4

**Theorem 4.2.** *For any $\alpha > 1$, $f_{\alpha-\text{Rényi}}$ is not weakly evaluable.*

*Proof.* Consider the following construction:

$$q_1(x) = \begin{cases} 1 - \eta - \eta e^{-M} & x = x_0 \\ \eta e^{-M} & x = x_1 \\ \eta & x = x_2 \end{cases}$$

and

$$q_2(x) = \begin{cases} 1 - \eta - \eta e^{-M} & x = x_0 \\ \eta & x = x_1 \\ \eta e^{-M} & x = x_2. \end{cases}$$

We have by symmetry that

$$f_{\alpha-\text{Rényi}}(q_1, q_2) = f_{\alpha-\text{Rényi}}(q_2, q_1) = \frac{1}{\alpha - 1} \log \left( (1 - \eta - \eta e^{-M}) + \eta e^{-\alpha M} + \eta e^{(\alpha-1)M} \right)$$

$$\geq \frac{1}{\alpha - 1} \log \eta e^{(\alpha-1)M}$$

$$\geq \frac{M}{2}.$$

The final inequality follows from choosing $\eta = e^{-\frac{(\alpha-1)M}{2}}$ and choosing $M \geq 2$ gives that $f_{\alpha-\text{Rényi}}(q_1, q_2) \geq 1$ (the same holds from $f_{\alpha-\text{Rényi}}(q_2, q_1)$ by symmetry of the construction).

Let $q^\star$ be chosen uniformly at random from $\{q_1, q_2\}$ and consider a dataset $S_{\text{eval}}$. Now, under either $q_1$ or $q_2$ the total mass on $\{x_1, x_2\}$ is less than $2\eta$. Thus, with probability at least $(1 - 2\eta)^m > 1 - 2\eta m$, $S_{\text{eval}}$ only contains $x_0$, in which the

distribution is identical under $q_1$ and $q_2$. It follows that any evaluation algorithm will rank the models incorrectly with probability at least

$$\frac{1}{2}(1 - 2\eta m),$$

which becomes arbitrarily close to $\frac{1}{2}$ as we increase $M$. □

**Corollary 4.3.** $f_{\mathrm{KL}}$ *is not weakly evaluable.*

*Proof.* We use the same construction as in the proof of Theorem 4.2. We have by symmetry that

$$f_{\mathrm{KL}}(q_1||q_2) = f_{\mathrm{KL}}(q_2||q_1) = \eta \log \frac{\eta}{\eta e^{-M}} + \eta e^{-M} \log \frac{\eta e^{-M}}{\eta} = \eta M(1 - e^{-M}).$$

Taking $M \geq \frac{2}{\eta}$ ensures that the above quantity is greater than 1. Let $q^\star$ be chosen uniformly at random from $\{q_1, q_2\}$. By the same argument as the proof of Theorem 4.2, $S_{\mathrm{eval}}$ will contain only $x_0$ with probability at least $1 - 2\eta m$, so any evaluation algorithm will misrank the models with probability at least

$$\frac{1}{2}(1 - 2\eta m),$$

which becomes arbitrarily close to $\frac{1}{2}$ as we decrease $\eta$. □

**Proposition 4.5.** *For any $N \geq 2$, $f_{N-\mathrm{Cov}}$ is not weakly evaluable.*

*Proof.* Fix any $c \geq 1$. Suppose for contradiction that $f_{N-\mathrm{Cov}}$ is $c$-weakly evaluable by some evaluation algorithm $\mathcal{A}$. Fix $\delta = \frac{1}{4}$ and choose $\gamma > 0$ sufficiently small so that $1 - \gamma > c\gamma + \frac{\gamma}{2}$. Set $\varepsilon = \frac{\gamma}{2}$ and let

$$m = m_{\mathrm{evl}, f_{N-\mathrm{Cov}}}(\varepsilon, \delta, c).$$

Choose $\eta \ll \gamma$ to be fixed later. Let $\mathcal{X} = \{x_0, x_1\}$ and consider the following construction:

$$q_1(x) = \begin{cases} \frac{1-\gamma}{N} & x = x_0 \\ 1 - \frac{1-\gamma}{N} & x = x_1 \end{cases}$$

and

$$q_2(x) = \begin{cases} 1 - \frac{\gamma}{N} & x = x_0 \\ \frac{\gamma}{N} & x = x_1 \end{cases}$$

and

$$q_3(x) = \begin{cases} 1 - \gamma & x = x_0 \\ \gamma & x = x_1 \end{cases}$$

and

$$q_4(x) = \begin{cases} 1 - \gamma - \eta & x = x_0 \\ \gamma + \eta & x = x_1 \end{cases}.$$

Now, under $q_3$, we have $f_{N-\mathrm{Cov}}(q_1, q_3) = 1 - \gamma$ and $f_{N-\mathrm{Cov}}(q_2, q_3) = \gamma$

It follows that

$$f_{N-\mathrm{Cov}}(q_1, q_3) = 1 - \gamma > c\gamma + \varepsilon = cf_{N-\mathrm{Cov}}(q_2, q_3) + \varepsilon,$$

so under $q_3$ as the ground truth, $\mathcal{A}$ must output $q_2$ with high probability. Otherwise, we would not have $c$-weak evaluability. Concretely,

$$\mathbb{P}_{S_{\mathrm{eval}} \sim q_3^m} [\mathcal{A}(\{q_1, q_2\}, S_{\mathrm{eval}}) = q_2] \geq 1 - \delta.$$

Similarly, under $q_4$, we have $f_{N-\mathrm{Cov}}(q_1, q_4) = 0$ and $f_{N-\mathrm{Cov}}(q_2, q_4) = \gamma + \eta$, so

$$f_{N-\text{Cov}}(q_2, q_4) = \gamma + \eta > \varepsilon = c f_{N-\text{Cov}}(q_1, q_4) + \varepsilon.$$

Therefore,

$$\mathbb{P}_{S_{\text{eval}} \sim q_4^m} \left[ \mathcal{A}(\{q_1, q_2\}, S_{\text{eval}}) = q_2 \right] \leq \delta.$$

Let

$$E = \{S_{\text{eval}} : \mathcal{A}(\{q_1, q_2\}, S_{\text{eval}}) = q_2\}.$$

Combining the two probability bounds above gives

$$q_3^m(E) - q_4^m(E) \geq 1 - 2\delta = \frac{1}{2}.$$

Thus,

$$\text{TV}(q_3^m, q_4^m) \geq \frac{1}{2}.$$

However,

$$\text{TV}(q_3, q_4) = \eta.$$

Choosing $0 < \eta < \min\left\{ \frac{\gamma}{2}, \frac{1}{4\max\{m,1\}} \right\}$ gives that

$$\text{TV}(q_3^m, q_4^m) \leq m\text{TV}(q_3, q_4) = m\eta < \frac{1}{4},$$

which is a contradiction. Hence $f_{N-\text{Cov}}$ is not $c$-weakly evaluable. The result follows by the fact that $c$ was chosen arbitrarily. $\square$

**Theorem 4.8.** *Fix $N \geq 2$, $\varepsilon, \delta \in (0,1)$, and any two candidate models $q_1, q_2 \in \mathcal{M}$ such that $(q^\star, q)$ have an $(N, \alpha)$ margin for some fixed $\alpha > 0$ for each $q \in \{q_1, q_2\}$. Then, there exists an evaluation algorithm $\mathcal{A}$ such that if $q^\star$ satisfies Assumption 4.6 for some fixed $\gamma > 0$, given evaluation data $S_{\text{eval}} = \{x_1, \ldots, x_m\}$ of size*

$$m \geq \max\left( \frac{3(N+\alpha)^2}{\alpha^2 \gamma} \log \frac{4}{\gamma\delta} \ , \ \frac{2}{\varepsilon^2} \log \frac{8}{\delta} \right),$$

*with $x_i \sim q^\star$, with probability at least $1 - \delta$,*

$$\mathcal{A}(\{q_1, q_2\}, S_{\text{eval}}) = q_1$$
$$\implies f_{N-\text{Cov}}(q_1, q^\star) \leq f_{N-\text{Cov}}(q_2, q^\star) + \varepsilon$$

*and*

$$\mathcal{A}(\{q_1, q_2\}, S_{\text{eval}}) = q_2$$
$$\implies f_{N-\text{Cov}}(q_2, q^\star) \leq f_{N-\text{Cov}}(q_1, q^\star) + \varepsilon$$

*Proof.* Fix $\varepsilon, \delta \in (0,1)$ and $q^\star, q_1, q_2$ that satisfy the assumptions in the theorem statement. Given a dataset $S_{\text{eval}} = \{x_1, ..., x_m\}$ with samples from $q^\star$, define the frequency distribution $\widehat{q}(x) = \frac{1}{m} \sum_{i=1}^m \mathbb{1}[x_i = x]$.

Now, define the score function

$$s(q, S_{\text{eval}}) = \sum_{x \in \text{supp}(\widehat{q})} \widehat{q}(x) \mathbb{1}\left[ \frac{\widehat{q}(x)}{q(x)} \geq N \right].$$

5

We ensure that with high probability,

$$\mathbb{1}\left[ \frac{\widehat{q}(x)}{q(x)} \geq N \right] = \mathbb{1}\left[ \frac{q^\star(x)}{q(x)} \geq N \right],$$

---

[5] If $q(x) = 0$, we adopt the convention that $\frac{\widehat{q}(x)}{q(x)} = \infty$. Intuitively, such a point should count towards the coverage profile penalty.

for each $q \in \{q_1, q_2\}$ and each $x \in \text{supp}(q^\star)$. In particular, we need that

$$\left| \frac{\widehat{q}(x)}{q(x)} - \frac{q^\star(x)}{q(x)} \right| < \left| \frac{q^\star(x)}{q(x)} - N \right|.$$

Equivalently, it suffices that

$$
\begin{aligned}
|\widehat{q}(x) - q^\star(x)| &< q(x) \left| \frac{q^\star(x)}{q(x)} - N \right| \\
&= q^\star(x) \left| 1 - \frac{N q(x)}{q^\star(x)} \right| \\
&\geq q^\star(x) \frac{\alpha}{N + \alpha},
\end{aligned}
$$

where the final inequality follows from Assumption 4.7. Thus, it is enough to require

$$|\widehat{q}(x) - q^\star(x)| < \frac{\alpha}{N + \alpha} q^\star(x).$$

Via the multiplicative Chernoff Bound, we get that

$$\mathbb{P}\left[ |\widehat{q}(x) - q^\star(x)| \geq \frac{\alpha}{N+\alpha} q^\star(x) \right] \leq 2 \exp\left\{ -\frac{m \alpha^2 q^\star(x)}{3(N+\alpha)^2} \right\} \leq 2 \exp\left\{ -\frac{m \alpha^2 \gamma}{3(N+\alpha)^2} \right\},$$

where the final inequality follows from Assumption 4.6. Now, union bounding over $x \in \text{supp}(q^\star)$ and applying Assumption 4.6 again gives

$$\mathbb{P}\left[ \exists x \in \text{supp}(q^\star) : |\widehat{q}(x) - q^\star(x)| \geq \frac{\alpha}{N+\alpha} q^\star(x) \right] \leq \frac{2}{\gamma} \exp\left\{ -\frac{m \alpha^2 \gamma}{3(N+\alpha)^2} \right\}.$$

Therefore, if

$$m \geq \frac{3(N+\alpha)^2}{\alpha^2 \gamma} \log \frac{4}{\gamma \delta},$$

then with probability at least $1 - \delta/2$, for every $x \in \text{supp}(q^\star)$ and every $q \in \{q_1, q_2\}$,

$$\mathbb{1}\left[ \frac{\widehat{q}(x)}{q(x)} \geq N \right] = \mathbb{1}\left[ \frac{q^\star(x)}{q(x)} \geq N \right].$$

Let $C(q) = \{x \in \mathcal{X} : \frac{q^\star(x)}{q(x)} \geq N\}$ denote the set of points that $q$ undercovers w.r.t $q^\star$. Under our earlier event, $s(q, S_{\text{eval}}) = \widehat{q}(C(q))$ and $f_{N-\text{Cov}}(q, q^\star) = q^\star(C(q))$. Applying a Hoeffding Bound and union bounding over $q \in \{q_1, q_2\}$, we have that with probability at least $1 - \delta/2$, given

$$m \geq \frac{2}{\varepsilon^2} \log \frac{8}{\delta}$$

samples,

$$|s(q, S_{\text{eval}}) - f_{N-\text{Cov}}(q, q^\star)| < \varepsilon/2$$

for each $q \in \{q_1, q_2\}$. From this, we get the following:

$$s(q_1, S_{\text{eval}}) \leq s(q_2, S_{\text{eval}}) \implies f_{N-\text{Cov}}(q_1, q^\star) \leq s(q_1, S_{\text{eval}}) + \varepsilon/2 \leq s(q_2, S_{\text{eval}}) + \varepsilon/2 \leq f_{N-\text{Cov}}(q_2, q^\star) + \varepsilon$$

and

$$s(q_2, S_{\text{eval}}) \leq s(q_1, S_{\text{eval}}) \implies f_{N-\text{Cov}}(q_2, q^\star) \leq s(q_2, S_{\text{eval}}) + \varepsilon/2 \leq s(q_1, S_{\text{eval}}) + \varepsilon/2 \leq f_{N-\text{Cov}}(q_1, q^\star) + \varepsilon.$$

Now, taking $\mathcal{A}$ to be the evaluation algorithm induced by $s$, the result follows immediately. $\qquad \square$

# C. Proofs From Section 5

**Theorem 5.3.** $f_{\mathrm{TV}}$ *is not weakly evaluable by the nll score. Additionally,* $f_{\beta-\mathrm{KL}}$ *is not weakly evaluable by nll score for any* $\beta \in \left(0, \frac{1}{2}\right)$.

*Proof.* Fix $c \geq 1$ and arbitrary $\varepsilon, \delta \in (0, 1)$.

Let $\mathcal{X} = \{x_0, x_1, x_2\}$ and fix $M > 0$ to be chosen later. Consider the following construction:

$$p := \frac{1 - \varepsilon}{4c}, \qquad r := \frac{1 + \varepsilon}{2} - \frac{p}{2},$$

$$q^\star(x) = \begin{cases} 1 - p & x = x_0 \\ p & x = x_1 \\ 0 & x = x_2 \end{cases},$$

$$q_1(x) = \begin{cases} 1 - p - r & x = x_0 \\ p & x = x_1 \\ r & x = x_2 \end{cases},$$

and

$$q_2(x) = \begin{cases} 1 - pe^{-M} & x = x_0 \\ pe^{-M} & x = x_1 \\ 0 & x = x_2 \end{cases}.$$

Now, observe that $\mathrm{TV}(q_1, q^\star) = r$ and $\mathrm{TV}(q_2, q^\star) = p(1 - e^{-M}) \leq p$. Hence

$$c\,\mathrm{TV}(q_2, q^\star) + \varepsilon \leq cp + \varepsilon = \frac{1 + 3\varepsilon}{4},$$

while

$$\mathrm{TV}(q_1, q^\star) = r \geq \frac{3 + 5\varepsilon}{8}.$$

Since $\frac{3 + 5\varepsilon}{8} > \frac{1 + 3\varepsilon}{4}$ for all $\varepsilon \in (0, 1)$, we obtain

$$\mathrm{TV}(q_1, q^\star) > c\,\mathrm{TV}(q_2, q^\star) + \varepsilon. \tag{1}$$

Consider $S_{\mathrm{eval}} = (x_1, \ldots, x_m) \sim (q^\star)^m$ and let

$$K = \sum_{i=1}^{m} \mathbb{1}[x_i = x_1],$$

so $K \sim \mathrm{Bin}(m, p)$. Now, via a simple calculation, we have that

$$\mathrm{nll}(q_2, S_{\mathrm{eval}}) - \mathrm{nll}(q_1, S_{\mathrm{eval}}) = \frac{m - K}{m} \log \frac{1 - p - r}{1 - pe^{-M}} + \frac{K}{m} \log \frac{p}{pe^{-M}} \geq \log(1 - p - r) + \frac{KM}{m}.$$

On the event $\{K \geq \frac{mp}{2}\}$, this quantity is positive provided

$$M > \frac{2|\log(1 - p - r)|}{p}.$$

Via a Chernoff bound, if $m \geq \frac{8}{p} \log(1/\delta)$ we have

$$\mathbb{P}\left[\mathrm{nll}(q_1, S) \leq \mathrm{nll}(q_2, S)\right] \geq 1 - \delta.$$

Combined with (1), it follows that $f_{\mathrm{TV}}(\cdot, q^\star)$ is not weakly evaluable by the nll score.

For the second part, fix $\beta \in (0, \frac{1}{2})$. Consider the following construction and fix $M > 0$ to be chosen later.

$$
q^\star(x) = \begin{cases} 1 - \beta & x = x_0 \\ \beta & x = x_1 \, , \end{cases}
$$

$$
q_1(x) = \begin{cases} (1 - \beta)e^{-M} & x = x_0 \\ 1 - (1 - \beta)e^{-M} & x = x_1 \, , \end{cases}
$$

and

$$
q_2(x) = \begin{cases} 1 & x = x_0 \\ 0 & x = x_1 \, . \end{cases}
$$

Observe that the only subsets on which $q^\star$ places at least $1 - \beta$ mass are $\{x_0\}$ and $\mathcal{X}$ itself. If we take $E = \{x_0\}$, then

$$
\mathrm{KL}_E(q^\star, q_2) = [\log(1 - \beta)]_+ = 0,
$$

so $f_{\beta-\mathrm{KL}}(q_2, q^\star) = 0$.

Now, we have

$$
\mathrm{KL}_{\{x_0\}}(q^\star, q_1) = \log \frac{1 - \beta}{(1 - \beta)e^{-M}} = M.
$$

Moreover,

$$
\begin{aligned}
\mathrm{KL}_{\mathcal{X}}(q^\star, q_1) &= (1 - \beta) \log \frac{1 - \beta}{(1 - \beta)e^{-M}} + \beta \log \frac{\beta}{1 - (1 - \beta)e^{-M}} \\
&= (1 - \beta)M + \beta \log \frac{\beta}{1 - (1 - \beta)e^{-M}} \\
&\geq (1 - \beta)M + \beta \log \beta.
\end{aligned}
$$

Choosing

$$
M > \max \left\{ 1, \frac{1 + \beta \log(1/\beta)}{1 - \beta} \right\}
$$

ensures that both $\mathrm{KL}_{\{x_0\}}(q^\star, q_1) > 1$ and $\mathrm{KL}_{\mathcal{X}}(q^\star, q_1) > 1$. Therefore,

$$
f_{\beta-\mathrm{KL}}(q_1, q^\star) > 1.
$$

Since $\varepsilon \in (0, 1)$ and $f_{\beta-\mathrm{KL}}(q_2, q^\star) = 0$, we have

$$
f_{\beta-\mathrm{KL}}(q_1, q^\star) > c f_{\beta-\mathrm{KL}}(q_2, q^\star) + \varepsilon.
$$

Take $S_{\mathrm{eval}} \sim (q^\star)^m$. Consider the event where there exists a sample from $S_{\mathrm{eval}}$ that is $x_1$. This occurs with probability

$$
1 - (1 - \beta)^m,
$$

which tends to 1 as $m$ tends to $\infty$. Thus, for $m$ large enough, this probability is greater than $\delta$. On this event, we have that $\mathrm{nll}(q_2, S_{\mathrm{eval}}) = \infty$ and $\mathrm{nll}(q_1, S_{\mathrm{eval}}) < \infty$. Thus, with probability greater than $\delta$, the nll score incorrectly ranks $q_1$ above $q_2$, failing our definition of evaluability. It follows that $f_{\beta-\mathrm{KL}}$ is not weakly evaluable by the nll score. $\qquad \square$

**Theorem 5.4.** $f_{\beta-\mathrm{KL}}$ *is not strongly evaluable.*

*Proof.* Fix an arbitrary sample size $m \in \mathbb{N}$. Consider the following construction:

Let $A$ be a finite set of size $N$, where $N$ will be chosen later, and let

$$
\mathcal{X} := A \cup \{x_1, x_2\}.
$$

Choose any $\gamma \in (0, \beta/2)$ and $a \in (0, 1 - \beta)$ to be fixed later, and define

$$
q_1(x) = \begin{cases} \frac{1-\beta}{N}, & x \in A, \\ \gamma, & x = x_1, \\ \beta - \gamma, & x = x_2. \end{cases}
$$

$$
q_2(x) = \begin{cases} \frac{a}{N}, & x \in A, \\ 1 - a, & x = x_1, \\ 0, & x = x_2, \end{cases}
$$

$$
q_0^\star(x) = \begin{cases} \frac{1-\beta}{N}, & x \in A, \\ \beta, & x = x_1, \\ 0, & x = x_2, \end{cases}
$$

and for each $S \subseteq A$ with $|S| = N/2$,

$$
q_S^\star(x) = \begin{cases} \frac{2(1-\beta)}{N}, & x \in S, \\ 0, & x \in A \setminus S, \\ \beta, & x = x_1, \\ 0, & x = x_2. \end{cases}
$$

First observe that $f_{\beta-\mathrm{KL}}(q_1, q_0^\star) = 0$ and $f_{\beta-\mathrm{KL}}(q_1, q_S^\star) = \log 2$ for all $S$.

Now define

$$
L_A(a) := \log \frac{1-\beta}{a}, \qquad L_x(a) := \log \frac{\beta}{1-a}.
$$

Since $a < 1 - \beta$, we have $L_x(a) < L_A(a)$.

Under $q_0^\star$, the ratio $q_0^\star(x)/q_2(x)$ is constant and equal to $e^{L_A(a)}$ on $A$, and constant and equal to $e^{L_x(a)}$ at $x_1$. Because $L_x(a) < L_A(a)$, the minimizing admissible set includes $x_1$ and then the smallest possible amount of $A$-mass needed to reach total $q_0^\star$-mass $1 - \beta$, namely $1 - 2\beta$. Note that since $A$ is finite, we may not achieve exactly a mass of $1 - \beta$. Thus, up to an $O(1/N)$ rounding error (which we can ignore for large enough $N$ to be selected later),

$$
f_{\beta\text{-KL}}(q_2, q_0^\star) = G_\beta(a) := \frac{(1 - 2\beta)L_A(a) + \beta L_x(a)}{1 - \beta},
$$

and

$$
f_{\beta\text{-KL}}(q_2, q_S^\star) = G_\beta(a) + \frac{1 - 2\beta}{1 - \beta} \log 2 + O(1/N).
$$

Now, in order to have the ranking of the candidate models under the ground truths flip, we need to fix $a$ such that

$$
0 < G_\beta(a) < \frac{\beta}{1 - \beta} \log 2 \tag{2}
$$

It is nontrivial that such an $a$ exists, but to see that it does, observe that $G_\beta(a)$ is continuous and tends to $\infty$ as $a$ approaches 0. Additionally, for sufficiently small $t$,

$$
G_\beta(1 - \beta - t) = \frac{-\beta}{(1-\beta)^2} t + O(t^2) < 0.
$$

It follows by the Intermediate Value Theorem that there exists $a \in (0, 1 - \beta)$ that causes (2) to hold.

Now, choose small $\eta > 0$ so that

$$
4\eta < \min \left\{ G_\beta(a), \frac{\beta}{1 - \beta} \log 2 - G_\beta(a) \right\}.
$$

Choose $N$ sufficiently large so that the rounding errors above are at most $\eta$ and so that

$$\mathrm{TV}\left((q_0^\star)^m,\ \mathbb{E}_S[(q_S^\star)^m]\right) < \frac{1}{2}.$$

This is possible because, for fixed $m$, the two sample laws differ by at most $O(m^2/N)$: marginally each draw has the same law under both distributions, and the only detectable difference comes from collisions among the $A$-samples.

By the choice of $\eta$, for every admissible $S$ we have

$$f_{\beta\text{-KL}}(q_1, q_0^\star) + 2\eta < f_{\beta\text{-KL}}(q_2, q_0^\star),$$

and

$$f_{\beta\text{-KL}}(q_2, q_S^\star) + 2\eta < f_{\beta\text{-KL}}(q_1, q_S^\star).$$

Fix

$$\varepsilon = 2\eta, \qquad \delta = \frac{1}{4}.$$

Assume for contradiction that $f_{\beta\text{-KL}}$ is strongly evaluable. Then there exists an evaluation algorithm $\mathcal{A}$ such that for every ground truth, with probability at least $3/4$, the algorithm selects the smaller-metric model whenever the metric gap exceeds $\varepsilon$.

Define the test

$$T(S_{\text{eval}}) := \mathbb{1}\{\mathcal{A}(\{q_1, q_2\}, S_{\text{eval}}) = q_1\}.$$

Under $q_0^\star$, $q_1$ is better than $q_2$ by more than $\varepsilon$, so

$$\Pr_{\mathcal{D}_0}[T = 1] \geq \frac{3}{4}.$$

Under every $q_S^\star$, $q_2$ is better than $q_1$ by more than $\varepsilon$, so

$$\Pr_{(q_S^\star)^m}[T = 1] \leq \frac{1}{4}.$$

Averaging over the random choice of $S$, we obtain

$$\Pr_{\mathcal{D}_1}[T = 1] \leq \frac{1}{4}.$$

Therefore

$$\mathrm{TV}((q_0^\star)^m,\ \mathbb{E}_S[(q_S^\star)^m]) \geq \Pr_{(q_0^\star)^m}[T = 1] - \Pr_{\mathbb{E}_S[(q_S^\star)^m]}[T = 1] \geq \frac{1}{2},$$

contradicting $\mathrm{TV}((q_0^\star)^m,\ \mathbb{E}_S[(q_S^\star)^m]) < 1/2$. $\qquad\square$

**Proposition 5.6.** *Fix $\varepsilon, \delta \in (0, 1)$ and two candidate models $q_1, q_2 \in \mathcal{M}$ such that $(q^\star, q)$ are $\Delta$-close in ratio for some $q \in \{q_1, q_2\}$ and $\Delta \in (0, \frac{1}{2})$. Then, if $\Delta \leq O(\varepsilon^2)$, there exists an evaluation algorithm $\mathcal{A}$ such that given evaluation data $S_{eval} = \{x_1, \ldots, x_m\}$ of size*

$$m \geq O\left(\frac{1}{\varepsilon^2} \log \frac{1}{\delta}\right),$$

*with $x_i \sim q^\star$, with probability at least $1 - \delta$,*

$$\mathcal{A}(\{q_1, q_2\}, S_{eval}) = q_1 \implies f_{\mathrm{TV}}(q_1, q^\star) \leq f_{\mathrm{TV}}(q_2, q^\star) + \varepsilon.$$

*and*

$$\mathcal{A}(\{q_1, q_2\}, S_{eval}) = q_2 \implies f_{\mathrm{TV}}(q_2, q^\star) \leq f_{\mathrm{TV}}(q_1, q^\star) + \varepsilon.$$

*Proof.* Fix $\varepsilon, \delta \in (0, 1)$ and models $q^\star, q_1, q_2$ that satisfy the assumption in the proposition statement. For any dataset $S_{\text{eval}} = \{x_1, ..., x_m\}$ sampled from $q^\star$, define

$$q_{\min}(S_{\text{eval}}) := \underset{q \in \{q_1, q_2\}}{\arg \min} \, \text{nll}(q, S_{\text{eval}})$$

and define $q_{\max}(S_{\text{eval}})$ analogously. Consider the evaluation algorithm $\mathcal{A}$ that outputs $q_{\min}(S_{\text{eval}})$. It suffices to prove that with probability at least $1 - \delta$,

$$f_{\text{TV}}(q_{\min}(S_{\text{eval}}), q^\star) \leq f_{\text{TV}}(q_{\max}(S_{\text{eval}}), q^\star) + \varepsilon.$$

Assume without loss of generality that $(q^\star, q_1)$ are $\Delta$-close in ratio. Then

$$\text{nll}(q_1, S_{\text{eval}}) - \text{nll}(q^\star, S_{\text{eval}}) = \frac{1}{m} \sum_{i=1}^{m} \log \frac{q^\star(x_i)}{q_1(x_i)}$$

is bounded. Thus, via a Hoeffding Bound, if $m \geq \Omega\left(\frac{1}{\varepsilon^2} \log \frac{1}{\delta}\right)$ and $\Delta \leq O(\varepsilon^2)$, then with probability at least $1 - \delta/2$,

$$\text{nll}(q_1, S_{\text{eval}}) \leq \text{nll}(q^\star, S_{\text{eval}}) + \Theta(\varepsilon^2).$$

That is, the nll score for $q_1$ must be close to the nll score for $q^\star$.

The next part of our proof is that if a model is far from $q^\star$ in TV distance, it is exponentially unlikely to have nll score close to $q^\star$. To see this, fix any model $q$ and define

$$L(q, S_{\text{eval}}) = \prod_{i=1}^{m} q(x_i),$$

the product-analog of the nll score. Rankings under nll are preserved under $L$. That is,

$$\sum_{i=1}^{m} \log \frac{q^\star(x_i)}{q(x_i)} \leq m\Theta(\varepsilon^2) \iff \frac{L(q, S_{\text{eval}})}{L(q^\star, S_{\text{eval}})} \geq e^{-m\Theta(\varepsilon^2)}.$$

Equivalently,

$$\sqrt{\frac{L(q, S_{\text{eval}})}{L(q^\star, S_{\text{eval}})}} \geq e^{\frac{-m\Theta(\varepsilon^2)}{2}}.$$

Applying Markov's Inequality, we get that

$$\Pr_{S_{\text{eval}}} \left[ \sqrt{\frac{L(q, S_{\text{eval}})}{L(q^\star, S_{\text{eval}})}} \geq e^{-\frac{m\Theta(\varepsilon^2)}{2}} \right] \leq e^{\frac{m\Theta(\varepsilon^2)}{2}} \mathbb{E}_{S_{\text{eval}}} \left[ \sqrt{\frac{L(q, S_{\text{eval}})}{L(q^\star, S_{\text{eval}})}} \right].$$

Now, we bound the expectation on the right-hand side as follows:

$$\mathbb{E}_{S_{\text{eval}}} \left[ \sqrt{\frac{L(q, S_{\text{eval}})}{L(q^\star, S_{\text{eval}})}} \right] = \mathbb{E}_{S_{\text{eval}}} \left[ \prod_{i=1}^{m} \sqrt{\frac{q(x_i)}{q^\star(x_i)}} \right] = \prod_{i=1}^{m} \mathbb{E}_{x \sim q^\star} \left[ \sqrt{\frac{q(x)}{q^\star(x)}} \right].$$

Therefore,

$$\mathbb{E}_{S_{\text{eval}}} \left[ \sqrt{\frac{L(q, S_{\text{eval}})}{L(q^\star, S_{\text{eval}})}} \right] = \left( \int_{\mathcal{X}} \sqrt{q^\star(x) q(x)} \, dx \right)^m = (1 - H^2(q^\star, q))^m \leq \exp\{-m H^2(q^\star, q)\},$$

where

$$H^2(q^\star, q) = 1 - \int_{\mathcal{X}} \sqrt{q^\star(x) q(x)} \, dx$$

is the squared Hellinger distance between $q^\star$ and $q$. Putting this all together gives that

$$\Pr_{S_{\text{eval}}} \left[ \text{nll}(q, S_{\text{eval}}) \leq \text{nll}(q^\star, S_{\text{eval}}) + \Theta(\varepsilon^2) \right] \leq \exp \left\{ \frac{m\Theta(\varepsilon^2)}{2} - m H^2(q^\star, q) \right\}.$$

Now, using the standard inequality

$$H^2(q^\star, q) \geq \frac{1}{2} \text{TV}(q^\star, q)^2,$$

we get that if $\text{TV}(q^\star, q) \geq \varepsilon$, then

$$\Pr_{S_{\text{eval}}} \left[ \text{nll}(q, S_{\text{eval}}) \leq \text{nll}(q^\star, S_{\text{eval}}) + \Theta(\varepsilon^2) \right] \leq \exp \left\{ \frac{m\Theta(\varepsilon^2)}{2} - \frac{m\varepsilon^2}{2} \right\}.$$

Taking the hidden constant in $\Theta(\varepsilon^2)$ sufficiently small and taking

$$m \geq \Omega \left( \frac{1}{\varepsilon^2} \log \frac{1}{\delta} \right),$$

we obtain that, with probability at least $1 - \delta/2$,

$$\text{nll}(q, S_{\text{eval}}) > \text{nll}(q^\star, S_{\text{eval}}) + \Theta(\varepsilon^2).$$

Now, to conclude the proof, consider $\text{nll}(q_{\min}, S_{\text{eval}})$. By definition, under the event from our Hoeffding bound,

$$\text{nll}(q_{\min}, S_{\text{eval}}) \leq \text{nll}(q_1, S_{\text{eval}}) \leq \text{nll}(q^\star, S_{\text{eval}}) + \Theta(\varepsilon^2).$$

By the argument above, any model $q$ with $\text{TV}(q^\star, q) \geq \varepsilon$ fails this inequality with probability at least $1 - \delta/2$. Hence, with probability at least $1 - \delta$,

$$\text{TV}(q^\star, q_{\min}) \leq \varepsilon.$$

Union bounding over $q_{\min} \in \{q_1, q_2\}$ gives that, if the nll score ranks $q_1$ above $q_2$, then $q_{\min} = q_1$, and so

$$f_{\text{TV}}(q_1, q^\star) = \text{TV}(q^\star, q_1) \leq \varepsilon \leq \text{TV}(q^\star, q_2) + \varepsilon = f_{\text{TV}}(q_2, q^\star) + \varepsilon.$$

Similarly, if the nll score ranks $q_2$ above $q_1$,

$$f_{\text{TV}}(q_2, q^\star) = \text{TV}(q^\star, q_2) \leq \varepsilon \leq \text{TV}(q^\star, q_1) + \varepsilon = f_{\text{TV}}(q_1, q^\star) + \varepsilon.$$

The result follows.

$\square$

