# OpenReview forum: "A Theoretical Framework for Statistical Evaluability of Generative Models"
_ICML.cc/2026/Conference — ICML 2026 regular_

### Official Review · Reviewer_7rXY · 2026-03-08

**Soundness:** 3
**Presentation:** 4
**Significance:** 2
**Originality:** 2
**Overall Recommendation:** 4
**Confidence:** 2

**Summary:**

This paper examines how different metrics used to evaluate generative language models (e.g., KL divergence, perplexity) using a finite iid test set. The paper introduces some formalism to help define this task.

First, the paper introduces the notion of "evaluability". This states that if a score function $s$ ranks model A as worse than model B on some finite test set $S_{eval}$, then with high probability, $f$ (the population level metric) also ranks the two models in the same way. The authors define c-weak evaluability when this holds up to some slack parameter $c$. Strong evaluability is when $c=1$. This is related to "estimability", which measures the convergence of $s$, a finite sample, to $f$.

The paper then gives some taxonomy of how various learning theory constructs (e.g., VC-dimension) relate to evaluability. For instance theorem 3.2 states that function classes with finite vc dimension are strongly evaluable. The authors later have results on other metrics, such as Renyi divergence and KL-divergence, showing that these metrics are not weakly evaluable.

**Compliance With Llm Reviewing Policy:**

Affirmed.

**Final Justification:**

I am maintaining my positive score, though I am admittedly less familiar with this area.

**Key Questions For Authors:**

Q1: How can you extend these results to multiple models?

Q2: The paper shows that NLL fails to evaluate TV distance. Can you clarify which metric perplexity is actually designed to evaluate, if any?

**Limitations:**

Yes.

**Strengths And Weaknesses:**

Strengths:

(S1) The paper is generally well-written and easy to follow.

(S2) The introduction of evaluability as a metric is novel and interesting. $c$-weak evaluability additionally adds some granularity to take this beyond standard estimability literature.

(S3) Theorem 3.2 is a very strong result.

Weaknesses:

(W1) All results presented in the paper are comparing just one model pair. However, a typical setting might include several pairs of models. One could perhaps still get similar guarantees as in the paper by union bounding over all ${k \choose 2}$ pairs, but this would be quite inefficient. What happens when you want guarantees over an entire set or ranking of models?

(W2) The non-evaluable metrics are informative, though not all that surprising. For instance, for KL-divergence the authors construct a pathological example where the two models disagree on a tiny mass, i.e., "For large enough M, x1 and x2 will never be observed in finite samples, rendering the two models indistinguishable by any score function." This is not a particularly strong result, as the distributions are pretty much the same where it matters (the large mass regions). Although this style of argument is common (e.g., see https://arxiv.org/pdf/2411.11824 theorem 4.5 as an example), I do not find it convincing as a reason to not use KL-divergence.

(W3) Although it is a theory paper, there are no experiments to empirically validate how often evaluability fails for some of these metrics in practice, for instance.

---

> ### Author Rebuttal · Authors · 2026-03-31
>
> We sincerely thank the reviewer for their time and feedback.
>
> > Q1: How can you extend these results to multiple models?
>
> We chose to define our framework in terms of pairwise comparisons between models because it keeps the definitions and results clean and independent of the size of the arbitrary candidate set. The negative results under this framework automatically extend to a framework in which we consider rankings of more than two models. The positive results extend neatly as well. There are two regimes:
> 1. In the strong evaluability (estimability) results for Theorem 3.2 and Proposition 3.4, the high probability event gives a uniform approximation of the population-level metric simultaneously for all models. Concretely, in the proof of Proposition 3,4, the bound $|s(q,S_{\text{eval}})-d_{\mathcal{F}}(q^\star,q)|\leq \epsilon/2$ depends only on the fact that $d_{\mathcal{F}}(q^\star,\hat{q})\leq \epsilon/2$.
> 2. For explicit guarantees over a fixed finite class of models $\mathcal{H}$, we would union bound over $|\mathcal{H}|\choose 2$ pairs, but the resulting sample complexity would only have logarithmic dependence on $|\mathcal{H}|$
>
> Additionally, even on an algorithmic level, there is a long line of work in hypothesis selection that studies the problem of, given $N$ candidates, outputing a distribution that is a close to an unknown distribution. Most recently, [Aamand et al, 2025](https://arxiv.org/pdf/2509.03734) have improved the proper case runtime to be near-linear in $N$.
>
> >Q2: The paper shows that NLL fails to evaluate TV distance. Can you clarify which metric perplexity is actually designed to evaluate, if any?
>
> Actually, we do not know of any reasonable evaluation metrics that perplexity evaluates without additional assumptions. While it is naturally associated with its population-level analog, KL divergence, our results show that perplexity does not weakly evaluate KL divergence or even $\beta$-Restricted KL divergence. KL divergence becomes unbounded if a candidate model misses a tiny bit of mass relative to the ground truth, and perplexity may not detect this in finite samples. Our discussion of perplexity is aimed at highlighting this idea: score functions need to be geometrically aligned with the metrics that they seeks to evaluate. For example, Proposition 5.5 says that TV distance is strongly evaluable by perplexity, when one of the candidate models and ground truth are close (under Assumption 5.4).

---

> > ### Author Rebuttal · Reviewer_7rXY · 2026-04-03
> >
> > Thank you for the comments. Given my recommendation was already an accept (4), I will maintain that score.

---

### Official Review · Reviewer_8N4s · 2026-03-08

**Soundness:** 4
**Presentation:** 3
**Significance:** 4
**Originality:** 3
**Overall Recommendation:** 4
**Confidence:** 4

**Summary:**

The paper introduces a new notion of evaluability. A metric $f:\mathcal{M}^{2}\to\mathbb{R}^+$ is $c$-evaluable if there is a score function $s:\mathcal{M}\times\mathcal{X}^{*} \to \mathbb{R}^{+}$ such that given a finite data $S$ of size $m$ dependent on $\epsilon, \delta$ and $c$ from $q$, w.p. $\geq 1-\delta$,

$$s(q_1, S) \leq s(q_2, S) \Rightarrow f(q_1, q) \leq c\cdot f(q_2, q) + \epsilon.$$

 Main contributions of the paper include study of evaluability of IMPs and Rényi divergences. The authors also show some example evaluation metrics that perplexity can or cannot evaluate.

**Compliance With Llm Reviewing Policy:**

Affirmed.

**Final Justification:**

The problem studied in the paper is novel and relevant. The authors provide some initial results.

**Key Questions For Authors:**

1. Do the authors know of any lower bounds for the sample complexities in Theorem 3.2 or Proposition 3.4? Is it possible to show lower bounds by the same dimensions or would it imply existence of finite taxonomy?

2. Could the authors clarify the definition of finite taxonomy and how Theorem 3.6 proves that it doesn’t exist? Because the theorem states functions $M_1, M_2,...$ don’t exist that works for all $\mathcal{F}$, but from the definition, these functions can depend on $\mathcal{F}$?

**Limitations:**

yes

**Strengths And Weaknesses:**

Soundness: The paper is technically solid, and the proofs seem to be correct.

Presentation: The presentation is clear and the paper is easy to follow. Perhaps, the related work section could be improved by expansion and adding more references to works studying density estimation and distribution learning as these problems are closely related to the evaluability problem.

Significance: I think the problem is relevant to the community and opens a new direction for future work.

Originality: The evaluation framework is novel and the authors do a good job of formalizing it. However, I think technical contributions are fairly limited as most positive results seem to follow using estimability and uniform convergence which are standard.

---

> ### Author Rebuttal · Authors · 2026-03-31
>
> We sincerely thank the reviewer for their time and comments. We appreciate the reviewer's suggestion that the Related Works section can be expanded and will include this in the final version.
>
>
> > Do the authors know of any lower bounds for the sample complexities in Theorem 3.2 or Proposition 3.4? Is it possible to show lower bounds by the same dimensions or would it imply existence of finite taxonomy?
>
> We thank the reviewer for this interesting question. The focus of Theorem 3.2 (and Proposition 3.4) is to establish the separation between strong and weak evaluability, rather than to optimize sample complexity. We agree that determining the optimal sample complexity is an important direction for future work. At present, it is unclear whether the same dimension characterizations yield tight lower bounds. However, the dependence on $\epsilon$ in our results is optimal.
>
> Regarding the final part of the question, proving lower bounds by the same dimensions would not contradict the result of Theorem 3.6. Theorem 3.6 rules out the existence of a fixed list of regimes that captures all estimable IPMs, but proving matching VC dimension lower bounds for the binary-valued IPMs in Theorem 3.2 would not contradict this. Additionally, the stated upper bounds in Proposition 3.4 are not in terms of a single parameter like VC dimension but rather $\text{PDim}_{\epsilon/24}$ is a function of $\epsilon$.
>
>
> > Could the authors clarify the definition of finite taxonomy and how Theorem 3.6 proves that it doesn’t exist? Because the theorem states functions F don’t exist that works for all , but from the definition, these functions can depend on?
>
> We want to thank the reviewer for pointing out an overload of notation that we will fix in the final version. To clarify, $\mathcal{F}$ in Definition 3.5 refers to a class of evaluation metrics, whereas $\mathcal{F}$ in Theorem 3.6 refers to the test class that underpins a particular IPM. The intent of Definition 3.5 is that the functions
> $M_1,M_2,...$ form a fixed universal list for the entire family of evaluation metrics. If the class of estimable IPMs were to admit a finite taxonomy, then for every $d_{\mathcal{F}}$ in the class, there is some index $i$ such that the sample complexity of estimating $d_{\mathcal{F}}(q_1,q_2)$ is at most $M_i(\epsilon,\delta)$ for sufficiently small $\epsilon,\delta$. That is, only the choice of $i$ depends on the particular IPM. Theorem 3.6 then shows that for any proposed universal list, one can construct a single estimable IPM whose estimation complexity is not upper bounded by any function in that list, so the set of estimable does not admit a finite taxonomy of sample complexities.

---

> > ### Author Rebuttal · Reviewer_8N4s · 2026-04-01
> >
> > Thank you for your response. I maintain my positive score.

---

### Official Review · Reviewer_u3fH · 2026-03-12

**Soundness:** 4
**Presentation:** 3
**Significance:** 3
**Originality:** 4
**Overall Recommendation:** 4
**Confidence:** 4

**Summary:**

The submission deals with an important and basic question:  can the generalization performance  of generative models be   evaluated from finite held-out samples?   The authors define   evaluation metric and  score function  in order to define  a notion of   evaluability of the metric.
Here the authors postulate the existence of a function m(epsilon,delta, c) such that if  the number of data samples, m, is larger than  m(epsilon,delta, c), then, if  the score function of  model q1   is smaller than the soce function of  model q2, both evaluated  evaluated at the the same held out  data, then the evaluation metric between q1  and the true distribution is smaller than the   the evaluation metric between q2 and  the true distribution.

The notion of estimability is studied with IPM  metric between a model and the true distrbution. IPM    is defined on   a class of test functions {\cal F} . If the test functions are binary and Vapnik-Chervonenkis dimension of {\cal F}  if finite, then    IPM  metric is evaluable, and the
function m(epsilon,delta, c) can be given explicitly (order O).  In this summary  the distinction  between strong and weak estimability  is neglected.  The authors provide a simular result for the case where   {\cal F}  consists of a bounded single function.

After this the authors prove that the well known Renyi alpha divergences are not evaluable, and a version of  the KL-divergence is also npt evaluable. The total variation distance is evaluable under certain conditions.

**Compliance With Llm Reviewing Policy:**

Affirmed.

**Key Questions For Authors:**

Q1: Limited evaluation.  Can  the results by evaluated by e.g. a real empirical example  or by simulation.
Q2: Are other phi-divergences evaluable?
Q3:  What about the Hellinger distance?

**Limitations:**

There is discussion of  potential negative societal impact of this work, and such a discussion  is probably not  relevant given the  nature of this work.

**Strengths And Weaknesses:**

The submission  includes only  theoretical results, and  the proofs are  correct  and often very standard.   The   results  are  based on reasonable assumptions.   The authors   discuss mainly    the strengths of their work.  The submission is clearly written and well structured.

The paper address an important  and  relevant problem   and  advances both  understanding   of  capabilities  and  practice in machine learning. The  submission  provides new insights  and  deepens  understanding   of existing methods.  It is not, of course,   clear  that
the theoretical set-up provided by this submission will be used by  every other researcher or practitioner.  This is a novel application of the Vapnik-Chervonenkis  theory  about finite samples.

---

> ### Author Rebuttal · Authors · 2026-03-31
>
> We sincerely thank the reviewer for their time and feedback.
>
> > Q1: Can the results by evaluated by e.g. a real empirical example or by simulation.
>
> This work is purely theoretical and serves as a first step toward a theory of statistical evaluation, where we formalize evaluability and establish initial results. We agree that developing empirical instantiations of this framework is an interesting direction for future work.
>
> > Q2: Are other phi-divergences evaluable?
>
> We believe that our negative results for the $\alpha-$Rényi divergences (and hence KL divergence by limit) extend to a broader subclass of tail-sensitive $\phi$-divergences. However, note that we do not have a general result for all $\phi$-divergences because the TV distance is simultaneously a $\phi$-divergence and an IPM. From Theorem 3.2, we know that TV distance is 3-weakly evaluable. It would be interesting to explore the characterization of evaluability for general $\phi$-divergences in the future work. We will add a discussion about this in the final version.
>
> > Q3: What about the Hellinger distance?
>
> For Hellinger distance, we can obtain a lower bound immediately from the same construction for binary IPMs in Theorem 3.2. Specifically, Hellinger distance itself is not $c$-weakly evaluable for any $c\lessapprox1.1233$.
> The upper bound is not immediately clear to us, but this would be an interesting direction for future work. We will add a remark after Theorem 3.2.

---

> > ### Author Rebuttal · Reviewer_u3fH · 2026-04-03
> >
> > I  feel satisfied with the replies to my questions and will not revise my overall evaluations.

---

### Decision · Program_Chairs · 2026-04-30

**Decision:**

Accept (regular)

**Comment:**

The submission introduces a formal theoretical framework for the statistical evaluability of generative models, defining the conditions under which a score function applied to a finite test set can reliably approximate a population-level evaluation metric. It distinguishes between strong evaluability and $c$-weak evaluability, showing that IPMs are evaluable for bounded test classes, while similarity-based metrics such as Renyi and KL-divergence are not, due to their sensitivity to rare events.

Reviewers agreed that the work was technically solid with correct, well-structured proofs. The introduction of the evaluability metric is novel and an interesting application of VC theory, with Theorem 3.2 as a notable highlight. I'm fond of the theoretical framework presented here, and there are a few nice takeaways. Although a few minor weaknesses were presented, all reviewers recommended acceptance. However, no reviewer was willing to extend beyond a weak accept, even though concerns were generally addressed. I suspect this is likely due to the more limited practical impact of the work, although this is likely the nature of this type of fundamental research.

I tentatively recommend acceptance, and suggest that the authors consider a possible approach to demonstrate this perceived limitation of KL-like metrics in a practical example in their next revision.